# Nanotechnology as a Promising Tool against Phytopathogens: A Futuristic Approach to Agriculture

Manjit Kumar Ray [1,†], Awdhesh Kumar Mishra [2,†], Yugal Kishore Mohanta [1,3,*,†], Saurov Mahanta [4], Ishani Chakrabartty [5], Neelam Amit Kungwani [6], Satya Kumar Avula [7], Jibanjyoti Panda [1] and Ramesh Namdeo Pudake [8,*]

1   Nano-Biotechnology and Translational Knowledge Laboratory, Department of Applied Biology, School of Biological Sciences, University of Science and Technology Meghalaya, Techno City, 9th Mile, Baridua 793101, India; manjit_ray2002@yahoo.com (M.K.R.); jibanjyotipanda83@gmail.com (J.P.)
2   Department of Biotechnology, Yeungnam University, Gyeongsan 38541, Republic of Korea; awadhesh.biotech07@gmail.com
3   Centre for Herbal Pharmacology and Environmental Sustainability, Chettinad Hospital and Research Institute, Chettinad Academy of Research and Education, Kelambakkam 603103, India
4   National Institute of Electronics and Information Technology (NIELIT), Guwahati Centre, Guwahati 781008, India; saurov.mahanta@gmail.com
5   Learning and Development Solutions, Indegene Pvt. Ltd., Manyata Tech Park, Nagarwara, Bangalore 560045, India
6   Marine Biotechnology, Gujarat Biotechnology University, Gandhinagar 382355, India
7   Natural and Medical Sciences Research Centre, University of Nizwa, Nizwa 616, Oman; chemisatya@unizwa.edu.om
8   Amity Institute of Nanotechnology, Amity University Uttar Pradesh, Noida 201313, India
*   Correspondence: ykmohanta@gmail.com (Y.K.M.); rnpudake@amity.edu (R.N.P.)
†   These authors contributed equally to this work and treated as co-first authors.

**Abstract:** It is crucial to increase agricultural yields to fulfill the rising demand for food and the security it provides for a growing population. To protect human food supplies and agricultural outputs, disease management is essential. Plant infections are a silent enemy of economic crop production and cross-border commerce of agricultural goods, inflicting roughly 20–30% losses a year. If infections are accurately and rapidly detected and identified, this can be minimized, and specialized treatment can be given. The current methods of preventing plant diseases are utterly dependent on agrochemicals, which have adverse effects on the ecosystem. By improving their solubility, lengthening their shelf life, and lowering their toxicity, nanotechnology can help reduce the harmful effects of pesticides and fungicides in a sustainable and environmentally responsible way. Engineered nanoparticles can be used to control plant diseases either by using the nanoparticle itself or as a carrier for fungicides and antibiotics. Regardless of the many prospective benefits of using nanoparticles, few nanoparticle-based products have been made commercially available for use in more widespread applications. For rapid and accurate spotting of plant diseases, the combination of nanotechnology systems with molecular diagnostics acts as an alternative where the detection may be taken in on a portable miniaturized appliance. By minimizing the application of chemicals and adopting quick identification of infections, nanotechnology might sustainably minimize many issues in disease control. This review outlines the tools and techniques used in the diagnosis of plant diseases and their management and explains how nanotechnology works, along with the current tools and their prospects for the future of plant protection.

**Keywords:** nanotechnology; sustainability; plant disease; detection; diagnosis

## 1. Introduction

By producing food and acting as a source of wealth for many nations, agriculture plays a crucial role in human development. Approximately 86% of rural residents rely on agriculture as their primary source of income [1]. Animal pests generate around 15–18% of agricultural losses, but weeds and microbiological illnesses represent 34% and 16% of crop losses, respectively [1,2]. Given that the predicted worldwide crop demand is rising day by day, sustainable intensification of agriculture is urgently needed [3]. This prognosis is troubling since agricultural productivity is the consequence of the combination of various abiotic and biotic factors. Abiotic stress can be brought on by adverse environmental conditions such as moisture, light, nutrient parameters, and the presence of harmful chemicals in the biosphere. Biologic stress is mostly brought on by infections with pathogenic microorganisms, such as those brought on by bacteria, fungi, viruses, and protozoa [4]. As per the recent prediction assessment of the UN Department of Economic and Social Affairs (UN DESA) report, the present global population will rise to 8.5 billion from 7.3 billion by 2030; in 2050, it will reach more than 9.5 billion; and in 2100, it will reach more than 11 billion. Therefore, crop output will need to improve to meet the demands of the fast-expanding global population, as crops are the key economic driver for a healthy and sustainable society [5]. It was estimated that globally, crop output suffers significantly from diseases and pests, with an annual loss ranging between 20 and 40% [6]. Different chemicals—pesticides, fungicides, insecticides, etc.—are frequently being used today for pest control measures. Despite their numerous benefits, such as high accessibility, speedy action, and reliability, pesticides have negative side effects on species that are not their intended targets, which can lead to the rebound of the pest community and the emergence of resistance [7]. Additionally, according to estimates [7,8], more than 80% of pesticides applied are irretrievable throughout or after administration. Hence, the development of cost-effective, ecologically acceptable, and highly effective pesticides is highly encouraged.

Current agricultural production practices make it difficult to achieve food security, according to recent figures on the world population. The extensive use of agrochemicals for crop protection and maximum agricultural output has a negative influence on the environment and causes a variety of health problems, some of which are even life-threatening for humans and other animals. Eutrophication and a considerable loss in soil fertility are further downsides of the current agrochemical-based farming system. Cutting-edge technology that can help with increased output and crop protection is urgently needed. The two most cutting-edge technologies that have been determined to have the ability to solve these major restrictions are nanotechnology and biotechnology [9]. As per the United States Environmental Protection Agency (USEPA), nanotechnology is the study of materials at sizes between 1 and 100 nm, where special physical characteristics allow for the development of innovative applications [10,11]. In the past two decades, research on science and technologies for agricultural and food systems that are enabled by nanotechnology has been started on a global platform. In the area of agriculture, nanoscience is helping to develop a variety of applications that are affordable [12]. Several of the biotechnological uses for nanoparticles have been foreseen in the past, including (a) the mitigation of issues with soil composition [13], soil salinity reclamation, and the stabilization of surfaces that are vulnerable to erosion [14]; (b) improving nutrient availability and mobility [15]; (c) observing environmental pollution [16]; (d) recognizing pH, moisture, and macronutrients in the soil [17,18]; and (e) delivering various agrochemicals, including pesticides [19], insecticides [20], herbicides [21], etc. Biosensors, barcoding combined with nanomaterials, antimicrobial food packaging, products that indicate the shelf life of agricultural commodities, pollutants and recalcitrant pesticides removal from water and soil and their bioremediation [22], and clay-based nano-constituents in water management are additional applications of nanotechnology in agriculture. In the last decade, antimicrobial nanoparticles have been used more often in agriculture [23,24]. Given the significance of agriculture, efforts are being made to preserve food security and sufficiency, and it is imperative to fully investigate the possibilities of nanomaterials in the control and diagnosis of

diseases and the genomic modification of plant disease resistance [4]. Due to their beneficial effects on plant development and resilience to biotic and abiotic stresses, nanomaterials have the potential to replace various agrochemicals in crop production. According to an earlier report [25], nanoparticles can also be utilized to address the problem of bacterial and fungal resistance to common fungicides and bactericides. Many different nanomaterials are now being researched for their effects and prospective uses, including metals and non-metals, polymers, carbon nanotubes (CNTs), quantum dots, etc. [26,27]. The management of phytopathogens at molecular levels and the detection of phytopathogens are unique capabilities of nanoparticles and have the potential to transform the food and agricultural industries. Pesticides can be loaded with nanomaterials to prevent photodegradation and allow for a controlled release of the pesticide, according to a recent report [28]. Similarly, semiconductor particles such as quantum dots may be utilized to create fluorescent markers for imaging at the cellular level [29]. The use of nanomaterials in health science is at an advanced stage and can be replicated in plants with similar success [30,31] (Figure 1).

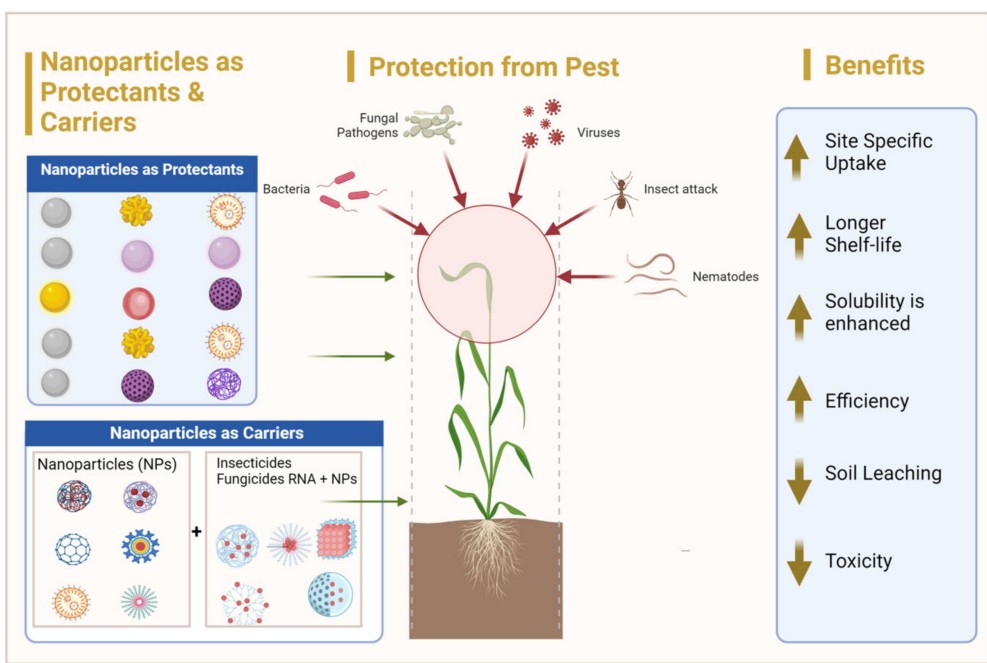

**Figure 1.** Application of nanoparticles in agriculture.

To safeguard agricultural goods from pre- and post-harvest illnesses, pathogen control is essential. It is still difficult to prevent these diseases, which are mostly brought on by bacteria, viruses, and fungi. The most popular method for controlling plant diseases is to take advantage of host-plant resistance. Conventional breeding involves raising and analyzing a sizable population of crops over several generations and calls for a supply of disease-resistant genes with optimal disease resistance [32,33]. To avoid the limitations of conventional breeding, transgenics can be a viable option [34]. Through the introduction of gene(s) from distant or unrelated species, transgenic plants display long-lasting resistance to one or more infections, which lowers the likelihood that the pathogen may acquire resistance [35]. However, the advantages of genetically modified crops have not yet been completely realized due to consumer concerns and international regulations regarding their safety [36]. The effects of overusing chemical pesticides on the environment are another problem that must be addressed. Nanotechnology is one of the alternatives that is being used more and more in this context. In nanotechnology, materials are modified at the atomic level to achieve special qualities that may be appropriately managed for the required purposes [37]. A nanoscale size regime is also where most natural activities occur. Therefore, the fusion of nanotechnology and biology has the potential to solve

many issues and transform the agricultural industry [38]. Nanoparticles with the required form, size, and surface features have been designed by material scientists accordingly, which can be utilized as shields or for accurate and targeted distribution of pesticides through different processes such as conjugation, encapsulation, and adsorption [39]. As agricultural nanotechnology improves, the ability to produce the genesis of insecticides and other active substances will significantly expand for plant disease management. One of two methods can be used to apply nanoparticles to protect plants: either they operate as crop protection agents on their own or they serve as carriers for active ingredients such as double-stranded RNA (dsRNA), which can be sprayed or soaked into roots, seeds, or foliar tissues. As carriers, nanoparticles can offer various advantages, including (i) longer shelf life, (ii) better pesticide solubility in water, (iii) minimizing toxic products, and (iv) more definite absorption into the target phytopathogen [40]. Nanocarriers may also increase the efficiency and endurance of nanopesticides in the midst of environmental stresses, allowing for fewer applications with lesser toxicity and lower costs [40,41] (Figure 2).

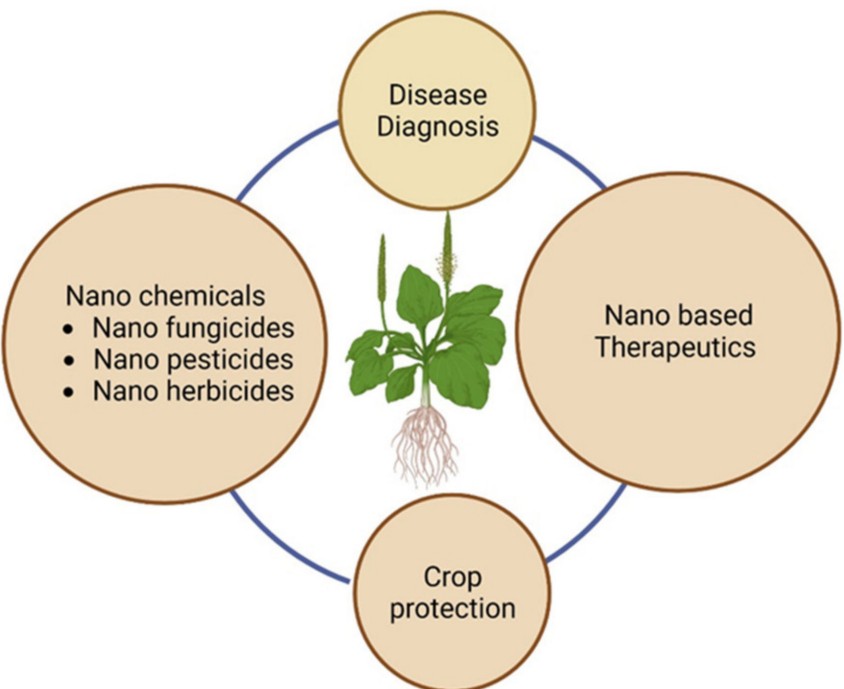

**Figure 2.** Nanotechnology-based strategy against phytopathogens.

## 2. Conventional Measures for Disease Control

One of the main causes of decreased productivity is plant disease. There are roughly 1000 diseases that affect the plants used to make economic crops and cause serious harm. Fungi account for most plant illnesses, followed by viruses, bacteria, nematodes, and several other plant pathogens. Serious plant diseases are also brought on by nutritional imbalances and vast environmental variations [42]. A traditional plant disease management operation includes good farming practices that will prevent subsequent infections, the eradication of infected plant tissues such as leaves and fruits to avoid pathogen transmission from the site of infection to other sites, and the application of chemicals such as pesticides, etc. for controlling insect vectors. Prevention strategies for disease transmission are more important than disease treatment when dealing with plant diseases. In general, integrated management techniques are chosen. In several crops, particularly horticulture, the adoption of plants with disease-resistant varieties and hybrids is strongly encouraged. With the advancement of technologies to limit the environmental application of harmful pesticides, genetically modified plant varieties are being created for disease-resistant traits. To lower the danger of plant diseases, producers also practice appropriate sanitation techniques,

including disinfecting tools and equipment. Plants should be protected against surface wounds that serve as entry points for diseases by using certified planting materials that are free of bacteria and viruses, rotating crops, and other preventative measures. Streptomycin usage has been reported since the 1950s, while copper has been utilized as an antibacterial component in agriculture for more than 130 years [43,44]. Utilizing bacteriophages specifically for the management of bacterial phytopathogens is an alternative strategy [4]. Crop rotation, sanitation practices, tillage, raised fields and beds, ridges, and mounds, as well as mulching, are other techniques [45]. Although there are many strategies for managing phytopathogens, there are drawbacks to each strategy that necessitate the adoption of two or more strategies to prevent crop losses brought on by diseases. Many nations had an abundance of agricultural products because of the "Green Revolution", but now there is a drop in food prosperity due to issues including climate change, declining quality of soil, lack of farming areas, expansion of inhabitants, and infections in plants and food. The creation of possible or compatible techniques that are authentic and invulnerable to the environment is urgently needed to attain food security. As a result, in the current environment, nanotechnology may be the latest tool to assist us in overcoming the difficulties of managing diseases while preserving the health of plants [4]. Due to technological advancements, societal demands, and budgetary limits, conventional agriculture, including disease management, is always changing. Improvements in integrated crop and pest management systems and the discovery and registration of low-risk fungicides have all contributed to the management of plant disease progression while preserving the environment. State-of-the-art disease detection and prevention in crops are crucial in minimizing disease-related damages to crops during growth and development, crop collocations, and post-harvest processing, along with gathering maximum yield and ensuring sustainable agriculture [46].

## 3. Plant Pathogen Detection with Nanotechnology

The primary reasons for limiting agricultural productivity are pathogen infections, which have become one of the key problems in the global scenario [47]. Despite being excessively slow and unsuited for general use, traditional methods for detecting pathogens and diagnosing plant diseases are typically only somewhat accurate. To identify plant pathogenic organisms with a high degree of accuracy and precision, traditional molecular diagnostic techniques, including polymerase chain reaction (PCR), enzyme-linked immunosorbent assay (ELISA), and other established techniques such as colony counting, fluorescence in situ hybridization (FISH), etc., are frequently used in laboratories around the world [48]. However, the use of these conventional methods in underdeveloped nations is constrained due to the demand for specialized equipment, laboratory setup, and experts to handle the equipment [49]. The application of nanotechnology in plant disease diagnostics might revolutionize research and lead to the development of cutting-edge instruments for the early and quick detection of plant infections. Nanomaterials are excellent options for this purpose because of their size (1–100 nm), which can offer improved surface-to-volume ratios and have exclusive chemical, photosensitive, and electrical properties that are not present in their bulk equivalents [50]. The ability to modify molecules at the nanoscale and the unique optical properties provided by nanomaterials will enable highly sensitive and useful detection of plant pathogens [4,51]. Biosensors, with the help of nanoparticles, can show better performance in selectivity, sensitivity, and detection limits. It is also possible to miniaturize the devices of various nanoparticles because of their small size. The nanoparticles provide advantages such as increased conductivity in the sensing platform; a high surface-to-volume ratio that increases the binding/immobilization surface for the bioreceptors; and tuning of the surface moieties on nanoparticles to create binding sites for biomolecule immobilization. Along with this, the use of molecularly imprinted polymers (MIPs) and DNA-based aptamers makes the biosensors more stable in all conditions. These properties make nanosensors a better alternative to conventional techniques [52]. Various applications of nanotechnology in plant disease detection are discussed herewith (Table 1, Figure 3).

**Table 1.** Nanotechnology-based diagnosis of plant pathogens/diseases.

| Nanomaterial/ Substrate | Disease/Causal Organism | Target Plants | Bio-Recognition | Detection, LOD/Accuracy | References |
|---|---|---|---|---|---|
| SWCNTs/Gold microelectrodes on a Si/SiO$_2$ wafer | Sec-delivered effector 1 (SDE1) | Citrus | Anti-SDE1 polyclonal | FET/LOD: 5 nM | [53] |
| Au and pentacene films/Gold gate electrode | PPV | Stone fruit trees | Anti-Plum Pox Virus polyclonal | EGOFET/LOD: 180 pg mL$^{-1}$ | [54] |
| SWCNTs/Silicon wafer covered with SiO$_2$ | p-Ethylphenol released by Phytophthora | Strawberry | ssDNA | E-nose 0.13% of Pethylphenol | [55] |
| N- and B-doped MWCNTs/Interdigitated Electroless nickel immersion gold electrodes | VOCs exhaled by Aspergillus and Rhizopus fungi | Strawberry | - | E-nose | [56] |
| rGO and Au NPs/ Kirigami-based structure with AgNW electrodes | VOCs exhaled by *Phytophthora infestans* infection | Tomato | | Chemiresistive sensor array/ >97% accuracy | [57] |
| Au NPs/GCE | *Xanthomonas axonopodis* | Citrus | Anti-PthA | FET-SWV/LOD: 0.01 nM | [58] |
| Au NPs/SPCE | CTV | Citrus | Thiolated ssDNA | EIS/LOD: 100 nM | [59] |
| Au NPs/SPCE | CTV detection | - | Thiolated primer | EIS L/OD: 1 pg μL$^{-1}$ | [60] |
| TiO$_2$ and SnO$_2$ nanoparticles/SPCE | p-ethylguaiacol, volatile compound due to *Phytophthora cactorum* fungus infection | | | CV and DPV/LOD: 35–62 nmol L$^{-1}$ | [61] |
| Au NPs/GCE | PSS | Stewartia | HRP | LSV/7.8 × 103 cfu mL$^{-1}$ | [62] |
| GO/Paper electrodes | False smut caused by *Ustilaginoidea virens* | Rice | ssDNA | CV and LSV/10 fmol L$^{-1}$ | [63] |
| GO/ITO | GBNV | Groundnut | anti-GBNV | DPV/LOD: 5.7 ng mL$^{-1}$ | [64] |
| Au NPs/SPCE | Detection of plant pathogen DNA | - | Recombinase polymerase amplification | DPV/214 pmol L$^{-1}$ | [65] |
| PPY nanoribbon/Gold microelectrode | Cucumber mosaic virus (CMV) | Cucumber | anti-CMV IgG | Chemiresistive Microelectrode/ LOD 10 ng ml$^{-1}$ | [66] |
| Au NPs/SPCE | CMV | Cucumber | - | Chronoamperometry | [67] |
| Au NPs/SPCE | Rice tungro disease | Rice | anti- RTBV/ RTSV | Cyclic voltammtry | [68] |
| Au NPs/Carbon ink 8-WE SPCE | CTV | Sweet orange trees | anti-bodies Ab1 and Ab2 | Amperometry/ LOD: 0.3 fg mL$^{-1}$ | [69] |
| Au NPs | *P. infestans* | Tomato | (Cys)-capped | C/0.4 ppm | [70] |
| Au NPs | *Xanthomonas campestris* | Brassica | | Colorimetric/102 CFU mL$^{-1}$ | [71] |
| Fluorescent nanoparticles | *Phakopsora Pachyrhizi* | Soybean | IgG antibodies | fluorescence 2.2 ng mL$^{-1}$ | [72] |
| Au NPs reverse primer (20-mer) | Yellow leaf curl virus | Tomato | Reverse primer (20- mer) | LSPR/5 ng μL$^{-1}$ | [73] |
| Au NPs-SA | *Alternaria panax Whetz* | Ginseng | Mouse anti-Fam antibody and BSA-Biotin | LFA/0.01 pg μL$^{-1}$ | [74] |
| Au NPs | *Phytophthora infestan* | Potato | Streptavidinbiotinylated T and C | LFA/0.01 pg μL$^{-1}$ | [75] |
| Au NPs and silver | Leafroll virus | Potato | Anti-PLRV antibodies | LFA 0.2 ng ml$^{-1}$ | [76] |
| Carbon nanoparticles | *X. arboricola pv. Pruni* | Stone fruits and almond | Polyclonal antibodies 2626.1-WC | LFA 104 CFU mL$^{-1}$ | [77] |

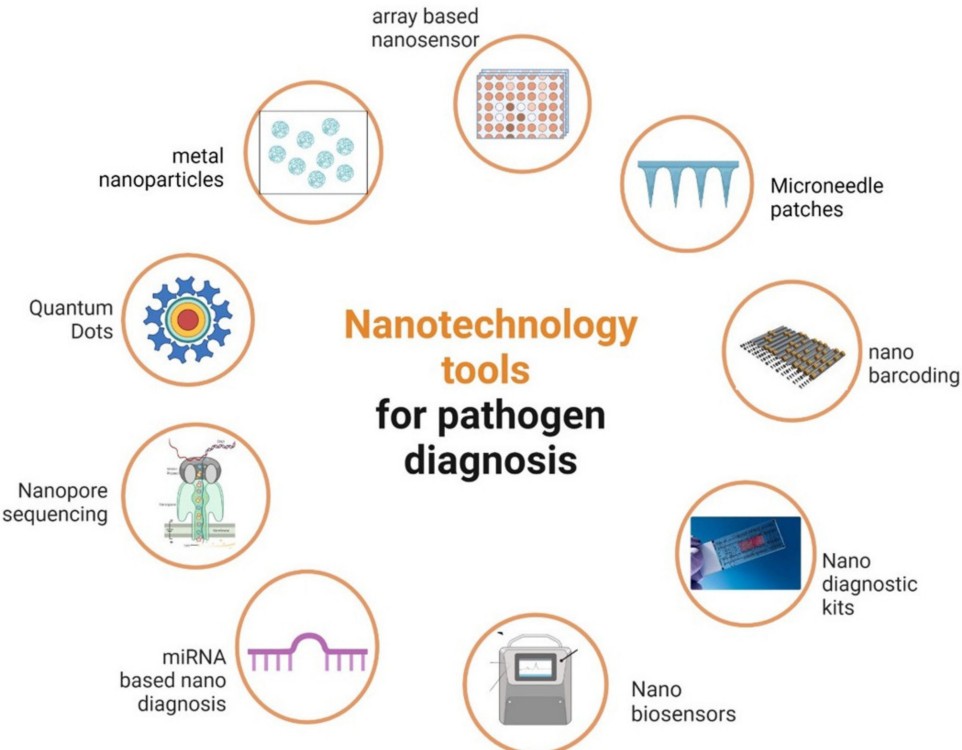

**Figure 3.** Diagrammatic representation of different nanotechnology tools for phytopathogen detection.

### 3.1. Bio-Nanomaterials

Bio-nanomaterials have greatly benefited the disciplines of biology, medical sciences, and agriculture. By creating more analytical tools in nanotechnology for accurate environmental hazard management, the use of bio-nanomaterials will be more effective in the environment [78]. In addition, pure cultures of microbes or their associated proteins and enzymes, as well as plant extracts, can be employed to bio-synthesize nanomaterials [79].

It has been demonstrated that nanotechnology is a key tool for identifying and quantifying plant pathogens. Examples of nanostructures with device miniaturization potential include those utilized to develop biosensors with increased selectivity, sensitivity, and limit of detection for onsite detection [55]. Sensor conductivity can be enhanced, for instance, by using carbon nanotubes or graphene derivatives [70,80]. Surface-functionalized electrospun nanofibers or metallic nanoparticles [81] significantly enhance the surface-to-volume ratio, allowing for bio-specific immobilization [58,70,82]. A molecularly imprinted polymer (MIP), which solely interacts with the target analyte and eliminates interferences, can be used to alter the sensor's selectivity [83].

Thermographic and hyperspectral imaging are some of the imaging techniques that have been used in the field for the indirect detection of plant disease. The sensitivity to changes in environmental parameters and the lack of specificity for different strains or disease subtypes are only two of its major shortcomings. Chemo- or biosensing technology has developed quickly, leading to a broad variety of useful applications, such as the assessment of the quality of pharmaceutical and industrial products. The analytes may be recognized by the nanosensors characteristic of electrical or optical outputs, which have recently been proposed and sold for agricultural diagnostics, depending on the transduction processes of the defined sensory interactions. The sensor's detection specificity may be improved by using selective chemical interactions or biospecies identification tools such as antibodies, enzymes, DNA oligos, and aptamers. To boost the detection sensitivity, surface-enhanced optical characteristics, such as electron-conductive nanoscale substrates or surface plasmon resonance (SPR), including graphene or carbon nanotubes, may be utilized as transducers [84].

### 3.2. Nanoparticle Bio-Barcode Assay

The bio-barcode test based on nanoparticles is more capable of identifying infections than standard techniques such as ELISA, real-time PCR, etc. [85]. It can also aid in the rapid recognition of plant diseases. In the bio-barcode method, magnetic microbeads (MMB) and gold nanoparticles (Au NP) are employed as probes. Many bio-barcode components have also undergone changes to broaden their use. DNA barcoding has been suggested as a method for detecting fungi DNA with a bio-barcode (b-DNA) [86]. For amplifying and locating proteins or nucleic acids, it is an exceptionally sensitive approach. In DNA bio-barcoded assays, the target protein is quickly separated from the sample using magnetic gold nanoparticles (Au-MNPs) that have been tailored with oligonucleotides for signal amplification. Considerable signal amplification is made possible by the high b-DNA-to-recognition agent ratio. It is particularly promising because, under ideal circumstances, it allows for the quick detection of a variety of nucleic acids at high zeptomolar levels and protein targets at low attomolar concentrations. A revolutionary approach that could replace the PCR technique is the bio-barcode test [50].

### 3.3. Nanopore DNA Sequencing

For disease management methods, accurate plant pathogen detection and identification are crucial. Common diagnostic techniques for identifying plant infections have drawbacks, including the need for previous knowledge of the genome sequence, limited sensitivity, and a constrained capacity to identify many diseases at once. The total nucleic acid concentration in biological samples may now be determined thanks to the advancement of DNA sequencing technology. Nanopore devices may be used to examine genetic data quickly and affordably without the requirement for sample preparation. Due to the protein nanopores in their membranes, this device distinguishes between nucleotides by measuring variations in conductivity. A bilayer membrane composed of polymers and nanopores in the chip is tied to a sensor. With the invention of this new technique, it is now feasible to recognize epidemics and track their progress, as well as differentiate between various pathogens, challenge genetic components, and sequence two different genes that are found on the same chromosome. By utilizing a nanopore model that is previously present in a contemporary diagnostic gadget, a whole genome analysis might be completed swiftly. To improve agricultural crops, it may be used to research the genomes of plants and diseases.

Plant pathogens pose a hazard to crop quality and productivity; as a result, effective and precise pathogen diagnostics are essential for managing and controlling crop disease. Research into plant viruses has been transformed by recent developments in sequencing technology. Because of its high sensitivity, high throughput, and lack of sequence dependency, next-generation sequencing (NGS), which represents metagenomics sequencing technology, has significantly advanced the development of viral diagnostics research. However, the expensive cost, labor-intensive nature, and cumbersome equipment of NGS-based viral identification techniques place a limit on their effectiveness. Long DNA or RNA readings may now be directly sequenced in real time, thanks to developments in nanopore sequencing. This is widely utilized in plant virus surveillance, virus discovery, viral genome assembly, and evolutionary research because of its versatility, portable sequencers, and adaptable data analysis [87]. Nanopore single-molecule sequencing technology is also being used to diagnose plant bacterial and fungal diseases. It was examined using DNA or RNA that had been obtained from the tissues of plants that had been injected with diseases that cause pathogens and exhibited the symptoms. Using nanopore sequencing, pathogens can be detected in real time and categorized to the species or genus level.

Conventional diagnostic techniques (including PCR, ELISA, and the Koch test) were used to validate DNA sequencing or direct RNA sequencing of samples containing unidentified disease pathogens, which corroborated the outcomes of nanopore sequencing. Long read lengths, quick run times, portability, cheap cost, and the potential for usage in any laboratory are all benefits of this technology [88]. The Oxford Nanopore Technologies tool

"MinION" is a handheld sequencing system, and it was found to be an efficient method for the diagnosis of various plant pathogens, including fungi such as *S. lycopersicum* in tomato and *P. digitatum* in lemon [88]. Two pathogens, *Candidatus Liberibacter asiaticus* and *Plum pox virus*, were quickly detected (within 24 h) in the peach by [89] using nanopore sequencing in conjunction with whole transcriptome amplification. By anticipating the existence of numerous plant viral species, such as *Dioscorea bacilliform virus*, *Yam mild mosaic virus*, and *Yam chlorotic necrosis virus*, in a water yam plant, [90] revealed high genome mapping findings attained by MinION. The entire experiment takes less than two hours, and the outcomes are equivalent to those of other diagnostic techniques (such as PCR and ELISA). Even though the present technique still has several drawbacks, such as a high per-read error rate and a limited ability to tell apart similar sequences, additional developments in nanopore technology will lead to the creation of more powerful sequencing platforms [84].

### 3.4. Nanodiagnostic Kit

A nanodiagnostic kit, often known as a "lab inside a box", is a small box that is used to monitor important plant processes that may be performed in a small space [91]. Several hurdles must be cleared before nanodiagnostic kit-based equipment systems may be used reliably in agriculture and related fields. The diagnostic kits' specificity may be improved, and strain differentiation can be achieved by several means, one of which is the identification and selection of efficient antigen, antibody, and nucleotide targets. It is also vital to create international standards for assessing tests and detection levels to compare studies on detection limits. Additionally, strategies for streamlining purification and isolating important genes are necessary for identifying the genetic targets of a particular illness [92].

These point-of-care kits and devices can assist farmers in limiting the spread of infectious illnesses by quickly identifying plant pathogens [93,94]. One strip with four mycosensors reportedly has the ability to detect ZEA, T-2/HT-2, DON, and FB1/FB2 mycotoxins in cash crops including wheat, barley, and maize [95]. Maize Chlorotic Mottle Virus (MCMV), the only member of the Mahromovirus genus, often co-infects plants with one or more viruses from the Potyvirus genus and presents a significant threat to the global maize economy. The application of viral integrated management techniques requires the swift and precise identification of the disease's causal agent. Six super-sensitive and precise monoclonal antibodies (mAbs) against MCMV were first developed in one study using pure MCMV virions as the immunogen. Following the discovery of the mAbs, the quantum dot enzyme-linked immunosorbent test (Dot-ELISA) was created, which was capable of detecting MCMV in the crude extract of infected maize leaf. A rapid and easy gold nanoparticle-based immunochromatographic test strip (Au NP-ICTS) based on the paired mAbs 7B12 and 17C4 was further developed to monitor MCMV in point-of-care testing. This test strip can identify the virus in crude extracts of MCMV-infected maize leaves that have been diluted 25,600 times ($w/v$, g/mL). It took 10 min to complete the whole ICTS test process. When compared to conventional reverse transcription-polymerase chain reaction (RT-PCR), the detection endpoint of both serological methods is higher than that of RT-PCR, notably the Dot-ELISA, which is 12.1 times more sensitive. Additionally, there was concordance between the RT-PCR outcomes and the detection outcomes of 20 blinded maize samples from the two serological assays. The newly created Dot-ELISA and Au NP-ICTS offer tremendous application potential for the detection of MCMV in plant samples [96]. Although nanotechnology has not yet been completely utilized to identify pathogen infections in agricultural crops, it has the ability to resolve many of the issues previously mentioned for efficient on-site real-time diagnosis of crop diseases [92].

### 3.5. Quantum Dots (QDs)

Semiconducting nanocrystals called quantum dots emit certain light wavelengths and change the exposed light spectrum into a distinct frequency of energy. They are three-dimensional nanoparticles with a broad excitation spectrum [97]. QD-based nanosensors

can be helpful for detecting a number of enzymes and infections [98,99]. Quantum dots (QDs) are made of elements from the periodic groups II–VI, III–V, or IV–VI that have special photophysical characteristics. They are also referred to as zero-dimensional materials since they are nanostructured materials. One of the most common QDs are cd-chalcogenide nanocrystals, which have a ZnS shell around a centrosome that is 2–10 nm in size. They typically range in size from 10–15 nm when the outside of the shell is covered with a polymer. According to reports, CdS, CdTe, and CdSe are typically employed as the centrosomes of quantum dots [100]. Other materials lack the unique photophysical properties of QDs. In contrast to conventional fluorescent probes such as fluorescent proteins and organic dyes, QDs are distinguished by their size-tunable light emissions, limited and symmetric emission spectra, and wide absorption spectra that provide simultaneous stimulation of various fluorescent hues. In addition, compared to other materials, QDs exhibit a remarkable increase in brightness and resistance to photobleaching [101–103]. Quantum dot-based biosensors make use of QDs as the interface component and have names such as QD-based BRET immunosensor, QD-based FRET immunosensor, and QD-based FRET genosensor, depending on the kind of molecular beacon attached to the QDs and transduction signals [104,105]. The conceptual basis of the QD-based FRET genosensor is commonly used in biological applications [99].

In one study, glutathione-S-transferase (GST) proteins, which are specific to Polymyxabetae, were detected using CdTe quantum dots coated with antibiotics as biosensors [106]. Fluorescence resonance energy transfer (FRET) depends on resonance dipole–dipole coupling, which is produced by rhodamine's interactions with CdTe quantum dots. In less than 30 min, this device may be used to evaluate plants and produce useful results. In a different investigation, *P. aurantifolia* was sensitively detected using a QD-based sensor, and the sensor's 100% specificity was demonstrated in sick lime plants [11]. Despite the fact that QD-based biosensors are a relatively new form of sensor and are predicted to provide new possibilities for managing plant diseases, some investigations have also been carried out on other agricultural pathogens by applying QD-based biosensors [107–109] for the detection of plant infections. It is plausible to assume that quantum dots will contribute to the impending revolution in plant pathogen detection if their unique photophysical properties are taken into consideration as an interface component [50].

## 4. Nanomaterials: Sustainable Weapons against Phytopathogens

Nanotechnology is currently being used more and more to create innovative antimicrobial compounds to control dangerous bacteria and fungi [110]. Because of their superior effectiveness against pathogens, nano-scale biocidal compounds play a pivotal role in modern medicine. For their antibacterial effect, metal nanoparticles' characteristics have been extensively researched (Figure 4). Nanoparticles with antibacterial action against a variety of microbes include alumina (Al), gold (Au), selenium (Se), silver (Ag), calcium oxide (CaO), copper oxide (CuO), magnesium oxide (MgO), silicon dioxide ($SiO_2$), titanium dioxide ($TiO_2$), and zinc oxide (ZnO) [111–113]. According to Lemire et al. [114], five broad mechanisms for the antibacterial activity of nanoparticles have been postulated for metal nanoparticles.

(1)  Damage to the membrane transporter and nutrient absorption systems.
(2)  Reactive oxygen species (ROS) production damages several cell organelles, including DNA, by inducing cellular and oxidative stress.
(3)  Toxic ion release causes changes in the permeability and activity of membrane proteins.
(4)  Cell death and genotoxicity are caused by the interrelationship of harmful ions produced by nanoparticles with DNA.
(5)  Energy production, membrane oxidation, and protein oxidation are all impacted by interference with metabolic processes. Depending on the size and dosage, the biocidal properties of nanoparticles are intended to provide distinctive and enhanced antibiotic activity [31].

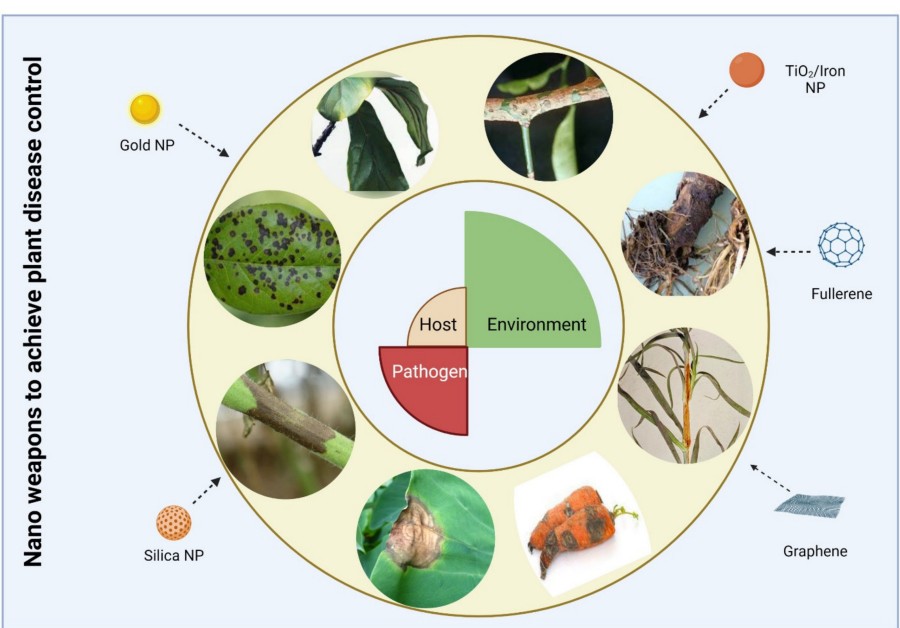

**Figure 4.** Application of different nanoparticles as nanoweapons to achieve plant disease control.

### 4.1. Nano-Fungicides

The development of nanoscale materials has advanced significantly in recent years, and they now exhibit unique properties that can make pesticide delivery safer [8]. When polymeric nano-fungicide formulations release agrochemicals slowly, they increase their bioavailability and have improved solubility [115,116]. The nano-fungicides created and examined so far have been successful in plant defense methods [117]. Nanoemulsions with smaller particle sizes, lesser viscosities, and greater stability for producing nano-fungicides [118] can be applied for plant protection. In a nanocapsule, the active antifungal component is contained within a core that is encircled by a membrane. Nanoencapsulation may also be used in the creation of nanopesticides. For crop protection, the potential use of polymers and inorganic substances in nanopesticide formulations has been investigated [119].

In the early twentieth century, insecticides were initially put into nanoparticles. Since then, experiments using diversified nanoparticles have included conventional pesticides. Silica, chitosan, and lipids were the three types of nanoparticle carriers that were most often studied. Researchers sought to increase water solubility and volatilization, intensify stability, and provide gradual release of the active compounds in these investigations. Low water solubility in pesticides necessitates the use of organic solvents, which raises their price and toxicity. Instead, the solubility may be improved using nanoparticles, which lessens the toxicity. Modified chitosan [120] and porous silica [121] have both been effectively used to load low-water-soluble pesticides.

### 4.2. Nanoparticles' Impact on Bacteria

According to recent studies, nanoparticles are capable of killing bacteria, which is mostly due to the breakdown of the bacterium's cell wall or the production of significant amounts of ROS [122–124]. For the treatment of bacterial diseases, antibiotics have been widely applied due to their cost-effectiveness and efficacious outcomes [125]. Excessive antibiotic usage results in the development of multidrug-resistant bacterial strains, as reported in several research studies. There is now an extremely potent type of bacterium that is resistant to antibiotics [126]. According to earlier studies, these bacteria include genes that are responsible for resistance [127]. Nanoparticles' direct interaction with the bacterial cell wall during their antibacterial action allows them to neutralize highly resistant bacteria [128,129].

## 5. Application of Various Nanoparticles against Plant Pathogens

### 5.1. Antimicrobial Activity of Metal Nanoparticles

Silver nanoparticles (Ag NPs) are the most promising (Figure 5) for antibacterial and antifungal action due to their unique structure, size, and flexibility based on electrical and optical activities. Fungicidal activity of Ag NPs was identified in *Aspergillus brasiliensis*, *Candida glabratus*, *C. tropicalis*, *C. krusei*, *C. albicans*, and *Penicillium oxalicum*. They alter membrane shape, leak cellular content, ATP, and cell membrane adhesion to fight pathogens [130]. Oxidizing lipids and proteins and mediating cellular and ROS toxicity; modifying the phosphotyrosine profile and modulating cell signaling; and damaging mitochondria are further methods [131]. Pathogens, including *Alternaria alternata*, *Cladosporium cucumerinum*, *Cylindrocarpon destructans*, *Didymella bryoniae*, *Glomerella cingulata*, *Botrytis cinerea*, *Corynespora cassiicola*, and *Fusarium solani*, were tested to see the effect of Ag NPs on their growth [132–134]. Additionally, Ag NP therapy greatly reduced the invasion of dangerous fungi such as *Magnaporthe grisea* and *Bipolaris sorokiniana*. *Macrophomina phaseolina*, *F. solani*, *Colletotrichum* sp., *Alternaria alternata*, *Enterobacter aerogenes*, *Klebsiella* sp., and *Bacillus cereus* were also tested for Ag NP susceptibility. In bactericidal tests, Ag NPs hindered phytopathogen DNA replication, produced ROS, and damaged bacterial DNA. Antibacterial Ag NPs interact with proteins to block enzymes, thanks to the thiol group in L-cysteine residues [135]. Vanti et al. [136] used *Gossypium hirsutum* extract to make 20–100 nm spherical Ag NPs, and these NPs were tested for their potential to suppress *Xanthomonas campestris* and *X. axonopodis* growth. Recent studies have found that *Ralstonia solanacearum* and *X. axonopodis* are inhibited by Ag NPs synthesized from *Solanum torvum* [135].

Along with fungal and bacterial pathogens, Ag NPs have also been found to reduce bean yellow mosaic virus (BYMV) concentrations, leaf lesions, and plant infection in *Vicia faba* [137]. They were also tested for tobacco mosaic virus (TMV) activity reduction in *N. tabacum* [138]. Mahfouza et al. [139] tested the virucidal effects of Ag NP (40–60 ppm) foliar spray after infecting plants with banana bunchy top virus (BBTV) [139]. The full suppression of SHRV illness in cluster beans treated with an aqueous solution of Ag NPs at 50 ppm demonstrates the efficacy of Ag NPs as an antiviral agent [140]. Using newly isolated strains of *Bacillus pumilus*, *B. persicus*, and *B. licheniformis*, Elbeshehy et al. [141] biosynthesized Ag NPs with in vitro-proven antiviral efficacy against BYMV infection. Similar results were seen when Ag NPs were administered 24 h after BYMV infection. El-Dougdoug & El-Dougdoug [142] found that the use of Ag NPs inhibited the spread of TMV and PVY in tomatoes. Ag NPs at 50 ppm increased TSP levels, peroxidase (POD), and PPO activity in tomatoes, resulting in SAR against TMV and PVY.

TMV- and PVY-infected tomato plants have fewer photosynthetic pigments and greater total soluble phenols and free proline [142]. Because viral signs were delayed, treated plants had less TYLCV disease severity than untreated controls.

Plant pathology nanobiotechnology research on Au NPs is rising due to their size, shape, regulated geometry, stability, energetic efficiency, dynamism, and safety during synthesis [143]. Plant disease research and treatment have extensively utilized their antibacterial properties (Figure 5). According to Hernandez-Diaz et al. [135], green-synthesized *Abelmoschus esculentus*-derived Au NPs exhibited promising antifungal activity against *A. niger* and *A. flavus*. Ag NPs and Au NPs fight plant viruses. In the case of yellow mosaic virus (YMV) and yellow dwarf virus (YDV) in barley, Au NPs eliminated virus-infected areas and the spread of pathogens [138]. Au NPs impede peptidoglycan formation, breaching bacterial cell walls and killing pathogens. DNA uncoiling and transcription can be inhibited by them too [135,144]. *Citrus sinensis* peel, *Azadirachta indica*, *Mentha spicata* leaves, and *Ocimum tenuiflorum* flower and leaf extract produce 20–30 nm gold nanoparticles, and these NPs inhibit *Pseudomonas aeruginosa* [145]. According to Payne et al. [146], Au NPs boosted Amoxicillin and Vancomycin's bactericidal properties.

Cu NPs manufactured using environmentally friendly methods could potentially manage the phytopathogens due to their high surface-to-volume ratio, which enhances

the pathogen interaction. In one study, nanochitosan increased Cu NPs' antifungal capabilities, and chitosan-coupled copper NPs (CS-Cu NPs) were very effective against *Pythium aphanidermatum*, *Trichoderma viride*, *A. flavus*, *Rhizoctonia solani*, *Fusarium moniliforme*, *F. oxysporum*, *Botrytis cinerea*, *Curvularia lunata*, and *Alternaria alternata* [147]. In another study, Cu NPs had antimicrobial effects at a wide range of concentrations, from 300 g/mL against *B. cinerea* in *Vitis vinefera* [148] to 250 mg/L against *Pythium ultimum* in *S. tuberosum* [149] and 300 ppm for black mold control in *Allium cepa*. After the green synthesis of 15-nm Cu NPs from *Syzygium aromaticum*, *Eugenia caryophyllata* showed antifungal efficacy against *A. niger* [150]. *Stachys lavandulifolia* and *Citrus medica* produced CuO, and Cu NPs were antibacterial and antifungal against *P. aeruginosa*, *Fusarium graminearum*, *F. culmorum*, and *F. oxysporium* (Figure 5) [151].

The many unique properties of nickel nanoparticles (Ni NPs) have made them a hot topic in recent years. Due to the small size effect, quantum size effect, and surface effect, they were excellent candidates for application as biosensors and inhibitory agents in plant pathology. They were found to be effective against CMV in tests conducted on *Cucumis sativa* [138]. There is a wide range of cellular functions that these NPs can disrupt, including ATP production, membrane permeability, and enzyme responses to environmental stress. The antimicrobial activity of Ni NPs has been demonstrated against a wide range of bacteria and fungi. Powerful antibacterial properties make it a possible alternative to synthetic agrochemicals that can be damaging to humans and the environment [138,152,153].

Se NPs have multiple applications in agriculture since they are less toxic than synthetic agrochemicals (Figure 5). *Emblica officinalis* fruit contained 15–40 nm Se NPs, according to research by Gunti et al. [154]. *Rhizopus stolonifer*, *Fusarium anthophilum*, *Aspergillus ochraceus*, *A. oryza*, *A. flavus*, and *A. brasiliensis* are all killed by Se NPs. *Trichoderma* sp.-extracted Se NPs at 200 ppm were evaluated for their efficacy against *Colletotrichum capsici*, a fungal disease of *Capsicum annuum*. There is some evidence that Se NPs can change microbial biofilms and display antifungal properties by inhibiting the germination of spores [155,156]. The antimicrobial actions of Se NPs are accomplished through regulation of intracellular ROS, disruption of target membranes, depolarization, and interference with metabolic interfaces by means of intracellular ATP concentration.

The most common targets of commercial uses of Si NPs against phytopathogens are *Magnaporthe grisea*, *Blumeria graminis*, *A. niger*, *Penicillium citrinum*, and *F. oxysporum* [138]. Coating Si NPs with antibacterial agents could cause membrane breakdown, ROS production, and cytotoxicity induction at varying doses. Although Si NPs have many beneficial effects, including improved seed germination, ion balance maintenance, metal ion absorption, reduced malondialdehyde levels, thicker cell walls, micronutrient transport, soil nitrogen regulation, and increased proline levels, Si NP accumulation in cells causes cellular damage [157].

Platinum nanoparticle (Pt NP)-based antibacterial drugs are promising, especially against bacteria [158]. Their catalytic activity, size, shape, and surface chemistry make them versatile. The Ag–Pt nanocomposite/polyaniline combination reduced *Streptococcus mutans* and *S. aureus* growth by activating antibacterial monomers [159]. Zhao et al. [160] found that 2–3 nm Au–Pt NPs composites have antibacterial activity against *P. aeruginosa*, *Salmonella choleraesuis*, *Klebsiella* sp., and *E. coli* (Figure 5).

Palladium (Pd) is an expensive metal with good electroanalytical, mechanical, and catalytic properties. The nanoscale structure suppresses bacterial development and interacts with biological particles. Pd NP complexes of polyamide S-rich sulfones were very effective against *S. aureus*, *E. coli*, *C. albicans*, and *A. flavus* [158].

Cadmium (Cd) is toxic to humans, animals, and plants [161]. *C. albicans*, *E. coli*, *S. aureus*, and *S. pyogenes* were investigated for the antibacterial activities of *Leucoena leucocephala* leaf extract-mediated synthesized CdO NPs and found these NPs very effective [162].

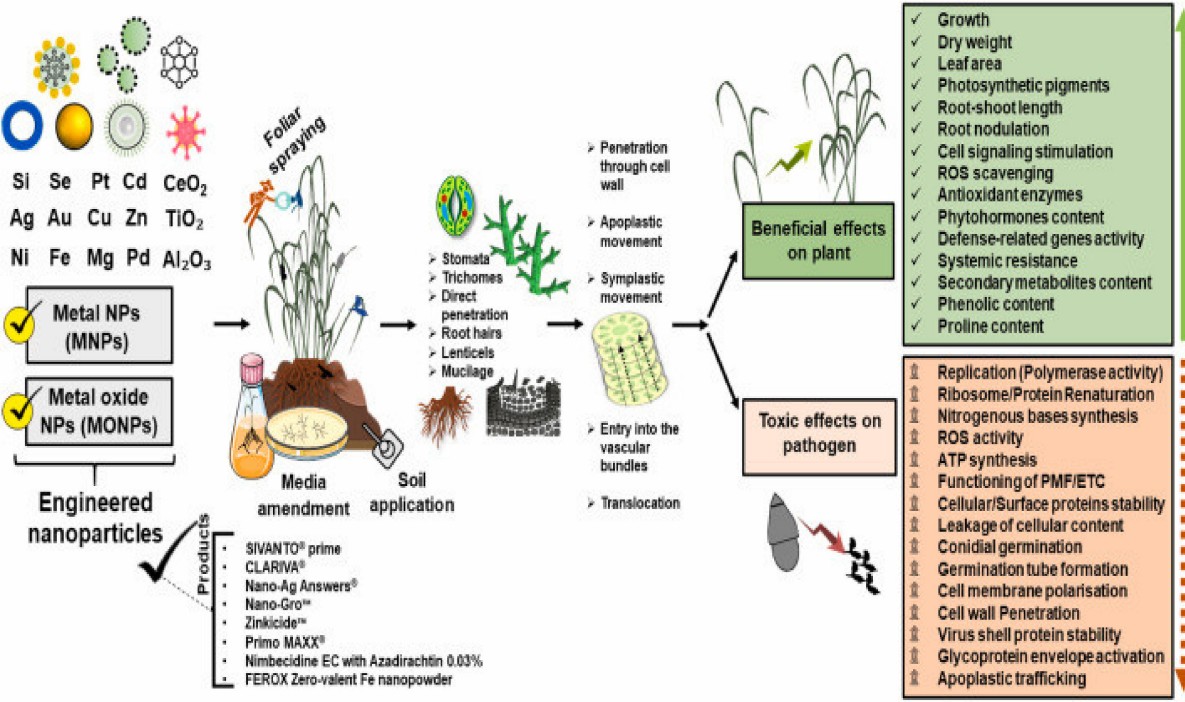

**Figure 5.** Application of metal and metal oxide nanoparticles in plant disease management, illustrating the positive effects (on plants) and inhibitory effects (on pathogens). Copyright permission: © 2022 Elsevier Ltd. All rights reserved. License number: 5618021061814 [163].

### 5.2. Antibacterial Activity of Metal Oxide Nanoparticles

Zinc (Zn) is used in the food, pharmaceutical, chemical, and agricultural industries. Due to their superior catalytic, optical, and physical qualities, ZnO NPs are the most frequently used in nanobiotechnology to make antimicrobial agents (Figure 5). The concentration-dependent micronutrient and pathogenic growth suppression of Zn make it a promising option for eco-friendly NP-based agrochemicals. Nano-Zn demonstrates inhibitory effects on various fungal and bacterial pathogens, including but not limited to *Sclerotinia sclerotiorum*, *Penicillium expansum*, *Rhizopus stolonifera*, *Rhizoctonia solani*, *Mucor plumbeus*, *Alternaria alternata*, *Fusarium oxysporum*, and *Botrytis cinerea* [164].

In addition to their demonstrated efficacy against fungi, ZnO NPs have also exhibited bactericidal properties against various bacterial strains. In a study conducted by Khan et al. [132], it was observed that ZnO NPs with sizes ranging from 12 to 100 nm exhibited bactericidal properties against various bacterial species, including *Staphylococcus aureus*, *S. pyogenes*, *S. epidermis*, *S. pyogenes*, *Enterococcus faecalis*, *E. faecalis*, *Bacillus subtilus*, *B. megaterium*, *P. aeruginosa*, *Sarcina lutea*, *Klebsiella pneumoniae*, and *Salmonella typhimurium*. The biofilms that have been stimulated and aggregated exhibit inhibitory effects on microbial growth, resulting in alterations to cell morphology and detrimental consequences for cellular integrity. Furthermore, the antimicrobial activity of ZnO NPs has shown a notable correlation with both particle size and concentration. Specifically, higher concentrations of ZnO NPs result in a larger surface area, which in turn enhances their antimicrobial effectiveness. Additionally, smaller-sized ZnO NPs exhibit greater ease in penetrating the bacterial membrane, owing to their higher interfacial area [132]. In a greenhouse study, ZnO NPs also increased brinjal plant CMV resistance, according to El-Sawy et al. [165].

In contrast to ZnO NPs, the utilization of CuO NPs has primarily been explored in the context of plant bacteria management. In a previously published study, it was observed that CuO NPs displayed the highest level of bactericidal activity against *E. coli*, followed by an inhibition against methicillin-resistant *Staphylococcus aureus* (SRSA) [166]. CuO NPs exhibit a notable capacity to regulate the pathogenicity of *Xanthomonas axonopodis*,

which is the causative agent responsible for the occurrence of blight in pomegranates [167]. Furthermore, Chen et al. [168] successfully produced CuO NPs by employing the extract from the *Carica papaya* plant, which exhibited notable bactericidal properties against the wilt pathogen known as *Ralstonia solanacearum*. The CuO NPs employed at a concentration of 150 g/mL exhibited a robust bactericidal impact, as reported by Mehrdad et al. [169] and Chen et al. [168]. According to predictions, copper (Cu) has the potential to selectively interact with sulfhydryl (-SH) groups present in crucial metabolic enzymes.

$TiO_2$ NPs, synthesized through the process of green synthesis, exhibited robust antimicrobial properties against various bacterial phytopathogens, including *Klebsiella pneumoniae*, *Staphylococcus aureus*, and *Proteus mirabilis* [170,171]. Additionally, $TiO_2$ NPs have high antagonistic properties against viruses. It restricts virus DNA replication in *Nicotiana benthamiana*, limiting the harmful activities of the Turnip Mosaic Virus (TuMV). In the targeted pathogen, $TiO_2$ NPs also activate ROS ($H_2O_2$ and $*OH$)-mediated cell wall and plasma membrane breakdown [172,173]. Green-synthesized $TiO_2$ NPs showed strong antimicrobial activity against *K. pneumoniae*, *S. aureus*, and *P. mirabilis*. $TiO_2$ NP-dsDNA biosensors utilizing surface functionalization tests may detect crown rot pathogens at 35 nm [174].

Aluminum (Al) is one of the most appealing, versatile, and economical metals. It is usually alloyed with other elements and has high thermal and electrical conductivity. Al is a commercial substance used in food, beverage, and structural engineering. In plant disease diagnosis and control, Al has shown promise [175,176]. $Al_2O_3$ NPs are a developing plant pathology tool that may control *Fusarium oxysporum*, *Pseudomonas aeruginosa*, and *C. elegans* [177]. $Al_2O_3$ NPs' positively charged surfaces stimulate electrostatic attraction with negatively charged cell membranes, causing adherence to the pathogenic surface and lowering cell viability. It also increases ROS production, causing membrane disruption, cell wall damage, and cell death [178]. Hyperaccumulation produces hydroxyl radicals in plants, which damage lipid, protein, and nucleic acid levels [172].

In addition, it has been demonstrated that silicon oxide nanoparticles ($SiO_2$ NPs) possess promising capabilities in the realm of disease management. $SiO_2$ NPs, with sizes ranging from 20 to 100 nm, exhibit inhibitory effects on the activities of the tomato yellow leaf curl virus (TYLCV). This leads to a decrease in the rate of pathogenesis and the concentration of TYLCV. Similarly, $SiO_2$ NPs with a size of 20 nm demonstrate comparable antiviral effects against Papaya ringspot virus (PRSV) and TMV, as reported by [138].

Iron oxide ($Fe_2O_3$), a micronutrient, is essential for plant metabolism, including respiration and photosynthesis. Devi et al. [179] found that 38 nm $Fe_2O_3$ NPs from *Platanus orientalis* leaves showed excellent antifungal efficacy against *Mucor piriformis* and *A. niger* at 0.1 mg/mL. Another study found that *Azadirachta indica* leaf extract produced $Fe_2O_3$ NPs that inhibited *Diplodia seriata*, *Botryosphaeria dothidea*, and *Alternaria mali* in apple orchids. $Fe_2O_3$ NPs also inhibited *A. niger*, *Alternaria alternata*, *Penicillium chrysogenum*, *Cladosporium herbarum*, and *Trichothecium roseum*, according to Parveen et al. [180]. Vargas-Hernandez et al. [138] evaluated 40–100 nm $Fe_2O_3$ NPs to reduce TuMV infection and replication in *Nicotiana tabacum*. *N. benthamiana* TMV is inhibited by 0.19 nm NPs.

As expected, $Fe_2O_3$ NPs' antibacterial effectiveness depends on cytoplasmic accumulation; hence, smaller NPs penetrate pathogen cell membranes more efficiently. It would cause NP accumulation and seepage of cellular components, which would increase the connection between cellular biomolecules and $Fe_2O_3$ NPs, causing DNA and protein structural alterations and bacterial cell death [181]. The green synthesis of $Fe_2O_3$ NPs produces many eco-friendly antibacterial chemicals that save energy and time. However, at large doses, $Fe_2O_3$ NPs cause growth deformation, spongy parenchyma cell deformation, abrupt chloroplast morphology, and a decrease in net photosynthetic rate [182].

A novel class of Mg-based NPs has outstanding mechanical, optical, chemical, and other properties. It can be used as a sensor, photonic device, adsorbent, or antibacterial because of its large surface area and reactive edge. The absorbent is particularly useful against microorganisms such as E. coli and S. aurens, and its characteristics improve as MgO size is decreased [152]. *Swertia chirayaita* extract is used to create environmentally

friendly MgO NPs, which are effective against *S. epidermidis*, *E. coli*, and *S. aurens* [183]. Saied and colleagues [153] employed agar-well diffusion to validate the antimicrobial's activity against *P. aeruginosa*, *B. subtilis*, *S. aureus*, and opportunistic yeast. Biogenic MgO NPs from *A. terreus* strains have shown antibacterial efficacy at 200 μg mL$^{-1}$ against pathogens such as *S. aureus*, *E. coli*, *B. subtilis*, *P. aeruginosa*, and *C. albicans*. MgO NPs inhibit plant-pathogen interactions by complexing MgO with pathogen cell walls, increasing ROS generation, alkalinizing microorganisms, and liberating Mg$^{2+}$. MgO NPs linked to lipopolysaccharides in bacterial cell membranes cause cell lysis. They also disrupt microbial quorum sensing, which stops the physiological process [153,183]. The FDA considers MgO NPs acceptable disinfectants due to their non-toxicity, biocompatibility, ease of availability, and environmental friendliness. MgO NPs promote growth and disease management, but they also produce phytotoxic injuries by damaging the cellular membrane. ROS content damages organelle membranes, causing cytoplasmic leakage and cellular inactivation [184].

Nanotechnologists are interested in cerium (Ce) because of its catalytic properties. Interestingly, CeO$_2$ NPs have antibacterial characteristics and could be used to regulate phytopathogenic activity. The photosynthesized NPs demonstrate promising antibacterial activities against *Pseudomonas aeruginosa*, *A. flavus*, *A. niger*, and *Fusarium solani*. Overall, CeO$_2$ NPs kill bacteria by interacting with bacterial membranes, blocking enzymatic activities, impairing cell respiration, oxidizing target organic materials, and adsorbing on bacterial surfaces through electrostatic interaction [138]. At higher doses, CeO$_2$ hyperactivates antioxidant enzymes and affects biomass production, leaf carbon buildup, photosynthesis, and oxidative stress [185]. The role and utilization of nanoparticles in controlling plant diseases caused by bacteria, fungi, and viruses are depicted in Table 2.

**Table 2.** The role and utilization of nanoparticles in controlling plant diseases caused by bacteria/fungi/viruses.

| Nanoparticles | Causal Organism | Disease | Target Plants | Sources |
|---|---|---|---|---|
| **Bacteria** | | | | |
| Ag | *Pectobacterium carotovorum* | Soft rot | *Beta vulgaris* | [186] |
| Ag | *Xanthomonas oryzae pv. oryzae* | Bacterial leaf streak | *Oryza sativa* | [187] |
| Ag | *X. oryzae pv. oryzae* | Rice bacterial blight | *O. sativa* | [188] |
| Ag | *Acidovorax oryzae* | BBS of rice | *O. sativa* | [189] |
| Ag | *Erwinia cacticida* | Soft rot erwinias | *Citrullus lanatus* | [190] |
| Ag | *A. oryzae* | BBS of rice | *O. sativa* | [191] |
| Ag | *Ralstonia solanacearum* | Bacterial wilt | *N. tabacum* | [192] |
| MgO | *Xanthomonas oryzae* | Bacterial blight disease in rice | *O. sativa* | [193] |
| TiO$_2$ | *Dickeya dadantii* | Bacterial root rot | *Ipomoea batatas* | [194] |
| TiO$_2$ | *Dickeya dadantii* | Soft rot | *Ipomoea batatas* | [195] |
| ZnO | *Xanthomonas oryzae* | Bacterial leaf blight diseases of rice | *O. sativa* | [196] |
| **Fungi** | | | | |
| Ag | *Sclerotium rolfsii* | Collar rot | *Cicer arietinum* | [197] |
| Ag | *Fusarium oxysporum* | Fusarium wilt | *Cicer arietinum* | [187] |
| Ag | *Phytophthora arenaria* | Crown and root rot | *Solanum lycopersicum* | [198] |

**Table 2.** *Cont.*

| Nanoparticles | Causal Organism | Disease | Target Plants | Sources |
|---|---|---|---|---|
| Ag | *Sclerotinia sclerotiorum* | Head rot | *Brassica oleracea var. capitata* | [199] |
| Ag | *Rhizoctonia solani* | Sheath blight | *O. sativa* | [200] |
| Ag | *S. sclerotiorum* | Gray mold | *Fragaria ananassa* | [201] |
| Ag | *Rhizopus stolonifer* | Discoloration of seed coat | *Hordeum vulgare* | [202] |
| Ag | *Phytophthora parasitica* | Black shank disease | *Citrus limon* | [203] |
| Ag | *F. oxysporum* | Fusarium wilt | *Gossypium hirsutism* | [204] |
| Ag | *F. oxysporum* | Black mold | *S. lycopersicum* | [205] |
| Au | *Puccinia graminis* | Wheat stem rust | *T. aestivum* | [206] |
| CS | *F. graminearum* | Seedling root rot | *Triticum aestivum* | [207] |
| Cu | *Aspergillus niger* | Black mold | *Allium cepa* | [208] |
| Cu | *Poria hypolateritia* | Red root | *Camellia sinensis* | [209] |
| Cu | *Penicillin digitatum* | Green rot | *Citrus sinensis* | [210] |
| Cu | *Botrytis cinerea* | Gray mold | *Vitis vinefera* | [148] |
| Cu | *A. niger* | Leaf rot | *Eugenia caryophyllata* | [150] |
| CuO | *Pythium ultimum* | Pink rot | *Solanum tuberosum* | [149] |
| CuO | *Rhizoctonia solani* | Root rot | *Solanum lycopersicum* | [211] |
| MgO | *A. oryzae* | BBS | *O. sativa* | [212] |
| MnO$_2$ | *A. oryzae* | BBS | *O. sativa* | [212] |
| Ni | *Colletotrichum musae* | Crown rot | *Musa acuminata* | [213] |
| Se | *Bipolaris sorokiniana* | Black point disease | *T. aestivum* | [214] |
| Se | *C. capsici* | Die back rot | *Capsicum annuum* | [215] |
| SiO$_2$ | *Aspergillus flavus* | Ear rot | *Zea mays* | [216] |
| Thiosemicarbazone NPs | *A. flavus* | Aspergillus ear rot | *Zea mays* | [217] |
| TiO$_2$ | *Bipolaris sorghicola* | Target leaf spot | *Sorghum bicolor* | [218] |
| TiO$_2$ | *B. sorokiniana* | Spot blotch disease | *T. aestivum* | [183] |
| TiO$_2$ | *B. sorokiniana* | Spot blotch disease | *T. aestivum* | [183] |
| ZnO | *F. oxysporum* | Vascular wilt | *Solanum lycopersicum* | [219] |
| ZnO | *F. culmorum* | Fusarium wilt | *Hordeum vulgare* | [220] |
| **Viruses** | | | | |
| Ag | SHRV | Chlorotic spot disease | *Cyamopsis tetragonoloba* | [140] |

**Table 2.** *Cont.*

| Nanoparticles | Causal Organism | Disease | Target Plants | Sources |
|---|---|---|---|---|
| Ag | BYMV | MD | *Vicia faba* | [141] |
| Ag | *Pyricularia grisea* | Rice blast | *O. sativa* | [221] |
| Ag | *Cucumis sativus* | Melon yellow spot | *Raphanus sativus* | [222] |
| Ag | TSWV | Spotted wilt disease | *S. tuberosum* | [223] |
| Ag | PVY | Potato tuber necrotic ringspot disease | *S. tuberosum* | [224] |
| Au | BYMV | MD | *Hordeum vulgare* | [225] |
| Au | BYMV | Yellow mosaic disease | *Hordeum vulgare* | [226] |
| CeO$_2$ | MTV | MD | *Datura stramonium* and *N. tabacum* | [227] |
| CNTs | MTV | MD | *N. benthamiana* | [228] |
| Fe$_3$O$_4$ | MTV | MD | *N. benthamiana* | [229] |
| GO-Ag | TBSV | Bushy stunt disease | *Lactuca sativa* | [230] |
| NiO | CMV | MD | *Cucumis sativus* | [231] |
| Schiff-based silver NPs | MTV | MD | *N. benthamiana* | [232] |
| SiO$_2$ | MTV | MD | *Nicotiana tabacum* | [229] |
| SiO$_2$ | TYLCV | Tomato leaf curl disease | *S. lycopersicum* | [224] |
| TiO$_2$ | BBSV | Mottle/mosaic disease | *Vicia faba* | [233] |
| ZnO | MTV | MD | *N. benthamiana* | [229] |

## 6. Action of Nanoparticles against Plant Pathogens—The Mechanism

Many factors have been identified as contributors to the toxicity of nanomaterials. Direct damage to the cellular membrane and interference with ATP generation and DNA replication are the results of ingesting free nano-ions. When millimolar concentrations of nano-ions are applied to unicellular organisms, the resulting morphological changes—including cytoplasm contraction, DNA condensation, and localization—are readily apparent (Figure 6), which allows the outflow of intracellular substances [234,235]. At very low concentrations, ions from NPs interact with respiratory enzymes such as NADH dehydrogenase and cause decoupling of respiration from ATP generation. Additionally, ionic nanoparticles and transport proteins attach, causing leakage of protons and a breakdown in the proton motive force [236]. Additionally, frequent DNA mutations were documented during gene polymerization in the PCR process and in *Escherichia coli*, where cells have been absorbing nanoparticles [237–244]. Nanotoxicity has been investigated in several biotic systems, including cell-line systems and various creatures, such as rats, aquatic animals, algae, and macrophages [245–252].

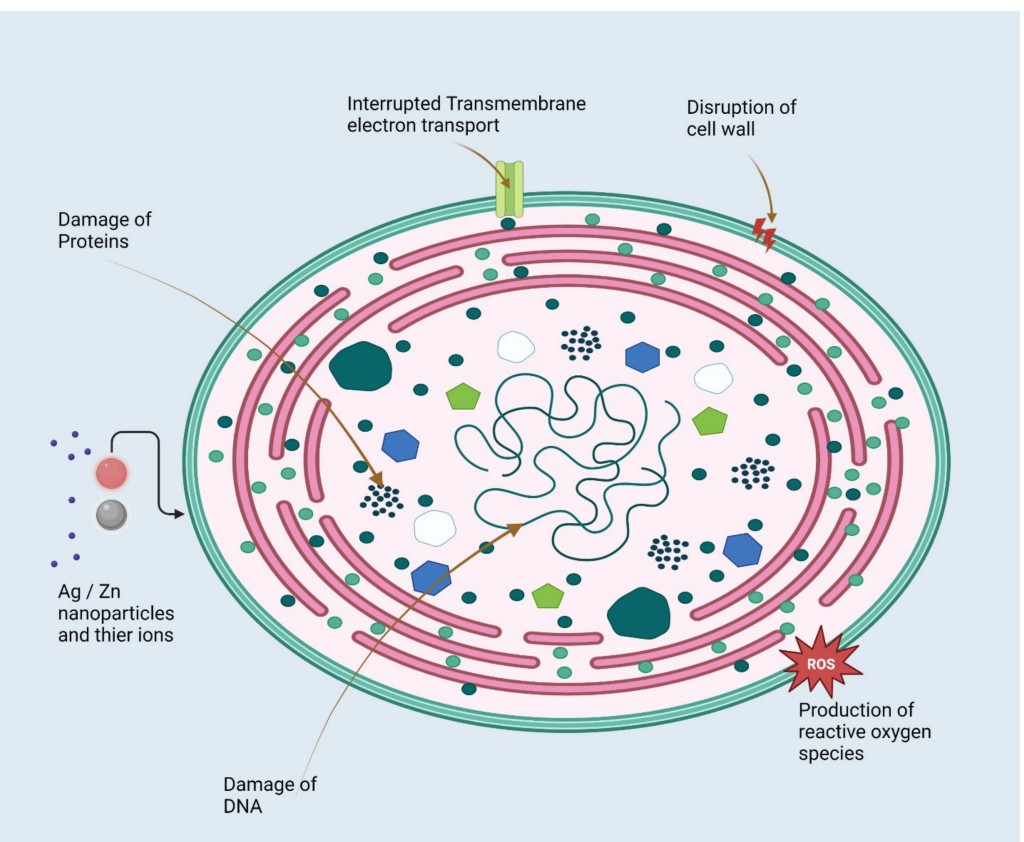

**Figure 6.** The possible mechanisms of nanoparticle toxicity against plant pathogens.

### 6.1. Formation of Reactive Oxygen Species (ROS)

Uptake of nanoparticles inside the cells causes the formation of ROS, which bring about oxidative stress and the development of nanotoxicity, which includes damage to DNA, uncontrolled signaling of cells, alterations in cell motility, cytotoxicity, and programmed cell death [245]. The chemical composition of nanomaterials determines the degree of ROS formation in targeted cells [239]. Inside the mitochondria, the reduction in molecular oxygen to water produces ATP through a series of linked proton and electron transport events. Superoxide anion radicals and later additional oxygen-containing radicals are created when a tiny portion of the oxygen is not fully reduced during this process. As a result, ROS are waste products of cellular oxidative metabolism, which largely takes place in the mitochondria. Some of the biologically significant ROS are hydroxyl radicals, superoxide anion radicals, hydrogen peroxide, singlet oxygen, etc. [246]. DNA is the primary cellular target of ROS. Moreover, crosslinks of DNA proteins, base and sugar lesions, single- and double-strand breaks, as well as basic site development, are all components of oxidative DNA damage [247]. While less reactive ROS may interact with DNA at a distance, more reactive radicals, such as hydroxyl radicals, may swiftly damage DNA in the area. Catalases, peroxidases, and superoxide dismutases (SODs) are a few of the well-known antioxidant enzymes that effectively guard against these damaging biotic processes. For instance, SOD catalyzes the conversion of hydrogen peroxide to superoxide. Superoxide has minimal reactivity toward most biological compounds and is a weak oxidant. The transformation of superoxide into a more reactive radical, notably the hydroxyl radical, is the cause of many harmful consequences for superoxide. Biological investigations have been based on the transformation of superoxide to hydroxyl or other additional strong oxidants [248].

### 6.2. Cell Membrane Damage

It was observed that when the cells are exposed to nanoparticles such as quantum dots, direct damage to the cell membrane takes place [249]. According to Akhtar et al. [250], the processes by which silica nanoparticles generated cytotoxicity and the ensuing oxidative stress in a dose-dependent manner were the formation of ROS and lipid peroxidation in the cell membrane. Furthermore, nano-CuO induces cytotoxicity in human alveolar epithelial cells, releases lactate dehydrogenase (LDH), and results in oxidative stress through the production of ROS and lipid peroxidation in a dose-dependent way. The specific mechanism by which nanoparticles interact with cell membranes and enter cells is not completely clear for all microorganisms. Electrostatic attraction between negatively charged cell membranes and positively charged nanoparticles is one concept that seems to explain the interaction between nanoparticles and cell membranes [251,252].

### 6.3. Liberation of Toxic Components

By releasing hazardous substances such as heavy metals or ions, certain nanoparticles cause toxicity in bacterial cells. Quantum dots (QDs) are semiconductor nanocrystals with a core composed of a noble or transition metal, such as Zn-Se, Pb-Se, Cd-Se, Cd-Te, Cd-Se-Te, or In-As, and a shell made of ZnS or CdS. They have an organic covering on top. *E. coli* and *Bacillus subtilis* have both been shown to take up the QD [253]. The toxicity of silver nanoparticles has been associated with the release of silver ions. The inactivation of vital enzymes is believed to be caused by the interaction of silver ions with protein thiol groups. Silver ions have also been shown to interfere with DNA replication and change the structure and permeability of the cell membrane [113,254]. Model organisms, including *E. coli*, *Staphylococcus aureus*, and *Candida* sp., have been used to study the antibacterial properties of nanoparticles [255,256]. Nanoparticles, or nanomaterials, are the active constituents of nanopesticides [117]. Nanomaterials and bio-composites were thought to be acceptable for use in the formulation of pesticides because they had specific characteristics, such as solubility, permeability, rigidity, thermal stability, crystallinity, and biodegradability [257,258]. These materials' increased affinity for the goal is due to their highly precise surface area [259]. A decrease in organic solvent flow-off and unwanted pesticide migration is achieved by nanoscale preparation of agricultural formulations, which increases their wettability and dispersion [117] and also contains nanoscale pesticide and insecticide distribution systems that exhibit controlled release properties [260]. Natural polymers such as cellulose, hemicellulose, albumin, gelatin, chitosan, sodium alginate, etc. can be utilized to create nanomaterials because of their non-toxicity, biodegradability, and affordability.

Nanoformulations that have been evaluated for disease control contain both inorganic nanomaterials and traditional fungicides. These formulations may boost disease control, reduce the use of harmful pesticides and their negative impacts on the ecosystem, and boost crop output [261,262]. It is necessary to formulate pesticides using nanomaterials to decrease the quantity of the active component while improving performance and to combat pesticide losses caused by evaporation and leaching. However, field applications will need safety and regulatory approvals following an ecotoxicological examination [263]. Recently, a study of the uses of nanomaterials as pesticides, micronutrients, fertilizers, and pesticide delivery agents was conducted [4,264] (Figure 6).

### 7. Environmental Risk of Nanoparticles

Nanomaterials have many uses, yet they have many adverse impacts. The enhanced surface activity and compact size of nanomaterials make them especially hazardous. They can easily penetrate cell walls and membranes to enter biological systems and stay there long enough to accomplish their functions. Nanomaterials can postpone or prolong toxicity effects and create various hard-to-predict effects, including neurotoxicity. Synthesizing metallic nanoparticles using physical and chemical methods is costly and risky since it requires the use of extremely reactive and hazardous reducing agents. Due to nanomaterials'

extensive use and benefits, research on their negative qualities and health risks is often postponed [265]. Stability can be a concern since "green" NPs tend to clump or dissolve in solution while being far less dangerous than chemically developed NPs. Modifying NP size, capping agents, and functionalization methods can modulate surface complexation processes, which affect NP stability [266]. Nanoparticle pollution poses serious risks to ecosystems; therefore, nanotechnology's rapid spread and use in several disciplines worry scientists. Since NPs are widely employed in biological sectors, including plants and agriculture, it is important to understand their detrimental impacts on human and environmental health. These materials solved several problems but caused others. Bioaccumulation of nanoparticles threatens humans and the environment [267]. Nanotoxicology, the study of negative effects and risks from nano-sized objects, has advanced in recent years. Nanotechnology has great promise due to its growing use in business, agriculture, medicine, and public health [268]. The development of nanotechnology is leading to more exposure to NPs in the ecosystem. While there is a wealth of data on NPs' impact across industries, the number of fatalities caused by metal-based NPs remains underreported.

In the health and welfare sectors, metallic NPs such as iron, silver, platinum, palladium, and gold and metal oxide NPs such as $Fe_3O_4$, $Fe_2O_3$, ZnO, and $TiO_2$ are useful. Several metallic NPs damage cell membranes, DNA, and proteins. These tiny NPs can also penetrate the bloodstream and harm key organs [269]. Metallic NP accumulation harms humans, plants, and crops. According to a 2014 survey, 2.7–3.1 lakh metric tons of NPs were generated globally in 2010 and were anticipated to reach around 5.9 lakh by 2019–2020 [266]. When compared to bulk chemicals, NPs absorb into any system by as much as fifteen to twenty times quicker due to their very small size. They get into the soil by several pathways and affect the natural flora and fauna there, such as plants that are good for the soil, bacteria, fungi, nematodes, and so on. Nanorelease is a danger when using NPs since it is not understood how the particles interact with their environment or how the weather (pH, salt concentration, etc.) affects them. Nanoparticles (NPs) made of silver, titanium, aluminum, zinc, nickel, indium, gold, copper, molybdenum, bismuth, iron, cobalt, silica, and tin are widely utilized in manufacturing. The most commonly produced and used metal-oxide NPs include ZnO, CuO, MgO, $TiO_2$, $Al_2O_3$, $SiO_2$, $Fe_2O_3$, $CeO_2$, $Cu_2O$, NiO, zirconium dioxide ($ZrO_2$), and lanthanum oxide ($La_2O_3$) [266,270].

Over the last five years, researchers have examined the influence of NPs on plants, bacteria, fungi, and soil nematodes. Soil features and complexity, such as buffering ability, organic nutrients, agglomeration and immobilization, accumulation, and environmental corona formation, influence how nanoparticles affect soil organisms [271]. Nanoparticles affect soil fertility, microbiology, and agricultural crops [272]. Nanoparticles' effects on soil microorganisms, especially those that benefit soil and plant health, must be studied. Soil and plant health depend on beneficial soil microorganisms, including bacteria and fungi [273,274]. Therefore, the subsequent section briefly covers the release of metal-based nanoparticles, their concentration in the environment, their interactions with various soil microbes such as plant growth-promoting rhizobacteria (PGPR) and fungi, and how these all relate to the nanoparticles' toxic effects on valuable soil microbiota.

### 7.1. Nanoparticle–Soil Microorganism Interaction

Soil microorganisms improve soil health by immobilizing nutrients, cycling carbon, and detoxifying pollutants [275,276]. About 15% of heterotrophic microflora are bacterial populations of various species [277] that can promote plant growth in a variety of ways. PGPR can colonize plant roots [278]. *Acinetobacter*, *Agrobacterium*, *Arthrobacter*, *Azospirillum*, *Azotobacter*, *Bacillus*, *Bradyrhizobium*, *Burkholderia*, *Pseudomonas*, *Rhizobium*, *Serratia*, *Thiobacillus*, and others are notable PGPR. Although diverse, only 2–5% of rhizosphere bacteria have powerful PGPR [279,280]. The relevance of PGPR to plant health makes NP–PGPR interactions vital [281]. Similar to other xenobiotics, the harmful effects of NPs on beneficial soil bacteria are emerging and still poorly understood. Due to the release of nanoparticle-based insecticides, fertilizers, and herbicides, NP–bacteria interactions must

be assessed. The direct entry of Fe NPs and TiO$_2$ NPs used in environmental cleanup and water treatment hinders target organism development. Fe NPs and TiO$_2$ NPs also harm non-target microorganisms and other living entities at the same concentrations. However, nZVI only harmed soil microbes [282]. The ZnO, CuO, Ag, FeO, and TiO$_2$ NPs showed varied chronic and acute toxic effects on pure microbial cultures and soil microorganisms. Size, surface charges, capping agent, divalent anions and cations, bacterial cell wall composition, and charge also affect the NP–bacteria interaction.

As pioneer colonizers, soil fungi breakdown dead plant tissues on and in the soil. Due to their mycelium network of branching, stiff tubes (hyphae) loaded with protoplasm, fungi naturally destroy dead materials. Thus, the fungal population and other soil organisms breakdown organic materials and provide plant nutrients. This role is crucial for crop pathogen protection. In agricultural and horticultural soils, arbuscular mycorrhizal fungi (AMF) are essential beneficial microorganisms that enhance root development, mineral cycling, ion uptake, and stress tolerance [283,284]. Additionally, antagonistic fungi such as *Trichoderma* sp. and *Glomus* sp. can reduce fungal infections to protect crops from plant diseases [285]. *Trichoderma* sp. (*T. asperellum*, *T. atroviride*, *T. harzianum*, *T. virens*, and *T. viride*) is often used in biostimulants and biocontrol formulations for horticultural crops [286]. Experiments with Ag, TiO$_2$, and ZnO NPs in soil have shown variable plant growth responses depending on NP type, size, and dose. Similarly, these NPs are recognized for their antibacterial properties against several bacteria and fungi. NPs enter fungal hyphae to distort and disrupt native shapes due to their size and nature. However, the NPs used to promote plant growth were contentious, so it is important to assess the effects of nanomaterials and nanoformulations on mycorrhizas and rhizobia. The interaction between NPs and mycorrhizal fungi affected its growth and had both positive and negative consequences, which are crucial for natural and agricultural ecosystem health, function, and sustainability [270]. Some NPs aid fungal colonization, whereas others hinder it. Therefore, understanding the mechanism of fungi–NP interaction is crucial.

### 7.2. Nanoparticles' Effect on the Bacterial and Fungal Populations in Soil

Nanomaterial-based sustainable agriculture relies on nanotechnology–agriculture compatibility. Sustainable farming might benefit from NP-based agro-chemicals and formulations, including nanofertilizer, nanopesticide, nanoherbicide, and nanosensors. These environmental NPs accumulate in the soil and impact native soil properties (Figure 7). Therefore, agri-nanotechnology's transport, bioavailability, and NP toxicity are typically cited as constraints. Agricultural scientists are trying to bridge gaps in their knowledge of agri-nanotechnologies by answering questions about how NPs interact with plants, soil, and soil biota.

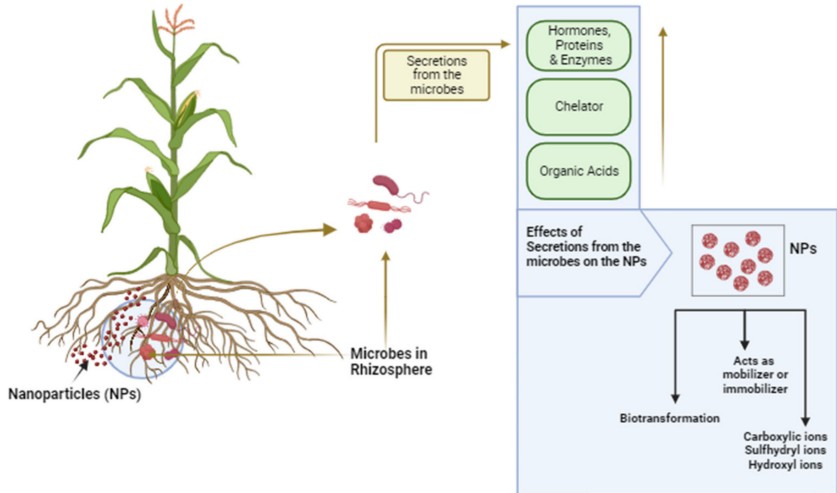

**Figure 7.** The diverse interplay among nanoparticles and soil microorganisms.

Over the past decade, numerous studies have examined how NPs affect soil microbial community structure [287,288]. In this perspective, a wide-ranging but interesting report showed that $TiO_2$ and ZnO NPs altered two significant soil bacterial communities: Rhizobiales, Bradyrhizobium, and Bradyrhizobiaceae (associated with nitrogen fixation) and Streptomycetaceae and Sphingomonadaceae (associated with organic pollutant and biopolymer decomposition). In particular, Ge et al. [289] found that $TiO_2$ and ZnO NPs reduced Rhizobiales, Bradyrhizobium, and Bradyrhizobiaceae and increased Sphingomonadaceae and Streptomycetaceae bacterial taxa dose-dependently. Later, ZnO NP-mediated toxicity negatively impacts soil microbial ammonification, respiration, and dehydrogenase activities [290]. Significant changes in dehydrogenase activity in bacteria (oligo and copiotrophs) and fungi were found in ZnO NP- and CuO NP-treated soil [291]. Similarly, the nitrate reductase activity of Azotobacter and Rhizobium was changed to 0.2 ppm due to Ag NP treatments [292]. Thus, different microbial species may respond differently to NPs. Several studies have shown that different mycorrhizal fungus species respond differently to NPs. Wang et al. [293] observed that *Glomus caledonium* could survive ZnO NP toxicity better than *G. versiforme*, affecting root colonization. This tolerance to heavy metals such as Zn, Cu, Pb, and Cd helped *G. caledonium* colonize further [293]. Even though there is clear evidence of NPs in soil microbial communities, there is a lack of literature linking soil variables to NPs' harmful behavior toward soil biota. The effect of different nanoparticles along with their concentrations is depicted in Table 3.

**Table 3.** Nanoparticles' impact on plant-friendly soil microbes.

| Nanoparticles | Concentration | Microbes | Types of Microbes | Impact | Sources |
|---|---|---|---|---|---|
| Ag | NR | *Av* | Bacteria | Biological nitrogen fixation was inhibited, ROS were produced, and cell number was reduced after exposure to NPs treated with Ag (size: 10 to 50 nm). | [294] |
| Ag NPs | 800 µg/kg sandy soil-loam mixture | *Ga* Faba bean | Fungi | Decreased glomalin levels, mycorrhizal responsiveness, and mycorrhizal colonization | [295] |
| Ag NPs | 12–36 mg/kg soil | AMF-Tomato | Fungi | Ag NPs reduce AMF colonization in a dose-dependent fashion. | [296] |
| Ag NPs and $Fe_2O_3$ NPs | 0.01–1 mg/kg Ag NPs and 0.032–3.2 mg/kg $Fe_2O_3$ NPs | *Tr* (Mycorrhizal clover) | Fungi | Mycorrhizal clover biomass drastically decreased, as did AMF's ability to absorb nutrients from its roots and its glomalin concentration. | [297] |
| Ag NPs and FeO NPs | Ag NPs (0.1–10 mg/kg) | Soil-microbial activity | Bacteria | FeO NPs had a favorable impact on the soil's microbial metabolism and nitrification capacity because they decreased the amount of ammonia-oxidizing bacteria in the soil and their abundance. | [298] |
| CuO NPs | | Nitrifying soil microbes | Bacteria | CuO NPs reduced the rate of nitrification. | [299] |

| Nanoparticles | Concentration | Microbes | Types of Microbes | Impact | Sources |
|---|---|---|---|---|---|
| CuO NPs | NR | Soil microorganisms engaged in C and N cycles | Bacteria | The amount of microbial activity related to C and N cycling was greatly reduced. CuO NPs' toxicity to microorganisms was not reduced by the experimental plant varieties, including wheat. | [300] |
| CuO NPs | | *Dv* (Sulfate reducing bacterium) | Bacteria | Sulfate reduction was inhibited by CuO NPs' catabolic and anabolic activity, while respiratory and electron transport genes were suppressed. | [301] |
| $MoO_3$ | 10 and 200 mg/L | *Af* and *An* | Fungi | Apoptosis was caused by metabolic alterations, changes in hyphae shape, and nuclear condensation, all brought on by exposure to NPs. | [302] |
| Nanodiamonds | 0.01–1 mg/mL | *Pc* | Fungi | Hyphal death, cell wall degradation, loss of cytoplasm, and oxidative stress-induced laccase and manganese peroxidase inactivation | [303] |
| Cirate-Ag NPs, bare-ZnO NPs, bare-CuO NPs, and bare-$TiO_2$ NPs | 100 mg/kg Ag NPs | Enzyme activity and the make-up of soil microbes | Bacteria | Particulate or dissolved application of Ag NPs inhibited specific soil enzymes, and the greater dose of Ag NPs altered the soil's microbial population. Enzyme activity was marginally suppressed by $TiO_2$ NPs. | [304] |
| PVP-coated Ag NPs | 10 and 100 μg/g Ag NPs | Ammonia-oxidizing bacteria | Bacteria | Soil nitrification and urease activities were drastically reduced. | [305] |
| $TiO_2$ NPs | 8, 16, and 33 mg/kg sandy soil | [a] AMF-Rice group | Fungi | Rice symbiosis with AMF is inhibited. | [306] |
| ZVI NPs | 50 mg/L | *Paracoccus* sp. | Bacteria | Increased cell proliferation and NO3 biodegradation; decreased cell density in a dose-dependent manner. | [307] |
| ZVS NPs: PVA and $Na_2$ ATP-doped | NR | *Ne* | Bacteria | Cell wall damage, nuclear fragmentation, oxidized NH3 capping, and size dependence | [308] |
| ZnO NPs | 30, 300, and 3000 ng/L | Marine and freshwater microcosms | Fungi | Affects the structure and activity of fungal communities and microorganisms negatively. | [309] |

**Table 3.** *Cont.*

| Nanoparticles | Concentration | Microbes | Types of Microbes | Impact | Sources |
|---|---|---|---|---|---|
| ZnO NPs | 6, 9, and 12 mmol/L | *Mc* and *Colletotrichum* sp. | Fungi | Fungal growth is inhibited by as much as 97% for M. citricolor and as much as 93% for *Colletotrichum* sp. | [310] |
| ZnO NPs | 500 mg/kg soil | *Fm*—Maize | Fungi | Negative impact on the AMF association | [311] |
| ZnO NPs | 800–3200 mg/kg loamy soil | *Gv*—Maize | Fungi | ZnO NPs disrupted the symbiosis of AMFs. | [312] |

NR: Not reported; AMF: Arbuscular mycorrhizal fungi; [a] AMF: Unspecified AMF species; *Av*: *Azotobacter Vinelandii*; *Ga*: *Glomus aggregatum*; *Tr*: *Trifolium repens*; *Ne*: *Nitrosomonas europaea* ATCC-19718; *Mc*: *Mycena citricolor*; *Fm*: *Funneliformis mosseae*; *Gv*: *Glomus versiforme/caledonium*; *Pc*: *Phanerochaete chrysosporium*; *Af*: *Aspergillus flavus*; *An*: *A. niger*; *Dv*: *Desulfovibrio vulgaris*; ZVI NPs: Zero-valent iron (Fe$^0$) nanoparticles; ZVS NPs: Zero-valent silver (Ag$^0$) nanoparticles.

## 8. Challenges and Limitations

Nanotechnology has the potential to revolutionize current pest management techniques and potentially offer answers for agricultural applications. The development of nanopesticides holds the promise of bringing about a number of never-before-seen advantages, such as: (i) improved solubility of pesticides; (ii) enhanced bioavailability and efficacy of pesticides when loaded onto nanoparticles with less toxicity; (iii) greater shelf life and organized target-specific supply of active components; (iv) pH-dependent release; (v) smart delivery of RNAi molecules for disease management; (vi) UV stability and rain-fastness with delaying in degradation of active components; and (vii) improvement in selective toxicity and combat pesticide resistance.

By virtue of their potential advantages for the environment and human health, nanopesticides are clearly a desirable development, as the earlier explanation makes clear. Agriculture nanotechnology, however, is not yet on the market. Since the bulk of the manufactured nanoparticle-based pesticides are still in the budding stages of development, further research is needed to determine the effectiveness and toxicity of the nanopesticides on soil and the environment. In terms of the use of pesticides, regulatory agencies have not provided a clear definition of what constitutes a nanopesticide. The effects of nanopesticides, in contrast to traditional pesticides, may depend on the bioavailability, concentration, absorption, and toxicity of the nanoparticles, as well as the ratio of the active constituents linked to them [313]. There is a dearth of knowledge on the issue of pesticide resistance and possible methods by which adding nanoparticles can reduce its prevalence [314]. Without the application of sophisticated analytical tools, it is impossible to develop regulatory criteria for risk assessment. Furthermore, the effect of increased nanomaterial manufacturing on ecosystem health has drawn some criticism. A sad reality remains that, to date, there are no defined protocols and regulatory standards for the use of nanomaterials, especially in the soil and aquatic ecosystems [266,315]. In addition, farmers and agricultural stakeholders may lack awareness and understanding of nanotechnology and its potential benefits. Effective knowledge transfer, education, and training programs are needed to bridge this knowledge gap and ensure the successful implementation of nanotechnology in agriculture. Furthermore, the successful integration of nanotechnology into existing agricultural practices can be challenging. Researchers and industries need to work together to develop scalable and practical solutions that can be easily adopted by farmers without disrupting their existing farming practices [266].

Addressing these challenges through research, collaboration, and responsible implementation is crucial for realizing the full potential of nanotechnology in agriculture and ensuring its practical or "on-field" approach. To prevent pesticide resistance, groups of pesticides must now be applied alternately, and future commercial uses will require a wide variety of nanopesticides. A number of factors, including the lack of knowledge on

the outcomes and safety characteristics of nanopesticides in long-term field trials, high manufacturing costs, the large amounts required, regulatory uncertainty, and the opinion of the public, need to be taken into account [26,316,317]. Receiving regulatory body clearance may be facilitated by using new tools and techniques to generate reliable data for analysis, characterization, and risk assessment. Material scientists and biologists must work closely together and bring in complementary talents from many fields to gain a deeper understanding of the fundamental interaction mechanisms in a complex bio–nano system. The rational choice of the most suitable nanoparticles may be aided by a complete understanding of the structural properties of the nanoparticles, including their shape, size, functional groups, and active adsorption/loading capacity. It is also critical to employ a reliable and repeatable approach in order to conduct biocompatibility and efficacy experiments at the cell, organism, and pest–host ecosystem levels under as-close-to-field conditions as is practical. Research on the potential of nanoparticles to produce useful products is now underway, which is encouraging for the future of agricultural nanotechnology research and development [41]. It is necessary to inform stakeholders of the ideal temperature for storing SLNs in order to address the problem of drug expulsion because the crystalline structure of the SLNs may result in drug expulsion due to the crystallization process under the storage conditions. When compared to neutral and anion species, nanoparticles (MSNs) with a positive charge on their surface can have a large cytotoxic impact. Therefore, this restriction might be removed by informing the stakeholders about the kind of nanoparticles they should purchase or provide to their clients. To avoid the problem of cytotoxicity, shareholders are suggested to always endeavor to favor the anion and neutral species of nanoparticles. Long-term interaction with or exposure to some metallic nanoparticles, such as silver nanoparticles (Ag NPs) and gold nanoparticles (Au NPs), has been found in some studies to have deleterious effects. This constraint might be resolved by notifying the shareholders to deliver the green or nanoparticles synthesized from natural materials to their clients in order to make the environment safe and to protect people, other plants, animals, and microbes from the risk of its toxicity. Biologists must work closely with material scientists and recruit the assistance of experts from other disciplines to gain a deeper understanding of the fundamental connections and mechanisms in a system of bio-nanotechnology. Conducting efficacy and biocompatibility studies at the organismal, pest–host environment, and cellular levels requires selecting a trustworthy and reproducible framework [2].

## 9. Existing Commercial Limitations

Achieving sustainable and environmentally friendly agricultural technology may eventually become a reality because of nanotechnology's promising outcomes in the agriculture sector, such as its novel technique of administering fertilizers, pesticides, and other materials [318]. According to recent research, earthworms are a helpful soil creature that might be harmed by nanomaterials [319]. Increased safety worries about nanoparticles in food and agriculture were summarized by the authors [320]. They focused on the most typical exposure pathways and contributing elements to nanotoxicity. Thanks to emerging technology, the environment is being exposed to an increasing number of man-made nanomaterials. The use of nanocarriers in agriculture is currently restricted by production scale and price. Costs will be significantly reduced by the large-scale manufacture of nanomaterials and their successful use in agriculture. Nanomaterials for agricultural uses face a challenging commercialization process that calls for well-protected materials, superior testing priorities, a precise risk assessment, and global regulatory guidelines [321]. Even if nanomaterials in bulk form are permitted for sale, many commercial nanomaterials are more harmful than their equivalents. The diverse uses of nanomaterials, including their manufacturing, toxicity, and utilization at the field level, still require more study.

## 10. Conclusions

Nanotechnology can improve disease resistance, nutrient use, plant growth, and controlled pesticide application. Pesticides, fungicides, insecticides, and herbicides may be utilized more efficiently and accurately by using environmentally friendly nanocapsules. Research and development in post-harvest nanotechnology is necessary to preserve quality and freshness while helping to prevent diseases. As the usage of nanotechnology increases, applications of green chemistry have reduced the need for hazardous solvents, enabling crop protection. Because of the utilization of biotechnology and nanotechnology, a far larger portion of the population can now safeguard and produce crops. Nanomaterials' effects on the environment are obvious, even though they are not yet established due to their unique physical and chemical properties. Given the novelty of using nanomaterials in agriculture, further research is required. The cost and environmental friendliness of crop protection systems are anticipated to be significantly impacted by nanomaterials. Utilizing nanotechnology will enhance disease diagnosis, enable molecular manipulation of both pathogens and plants, and facilitate the construction of cutting-edge methods for disease control in greenhouses and fields. As more research is devoted to finding, altering, and employing nanotechnology, we believe that the barriers to the global food supply will be eliminated. Now, only a limited number of laboratories are looking into the use of nanotechnology in phytopathology [11]. The study of plant genomics and gene functions can advance with the use of nanotechnology and nanoparticles in agricultural research. Nanoparticles can be utilized to transmit genes to plants, create disease-resistant plants, and enhance crop species. In comparison to viral nucleotide delivery by virus-induced gene silencing, nanoparticles designed with nucleotides will offer various benefits. Future crop improvement and pest and disease resistance might be achieved by nanoparticle-mediated gene delivery. The application of designed nanoparticles in the future may be for the clever delivery of nucleotides to plants, such as siRNAs. The nucleotides in these nanoparticles can be exploited to engineer resistance to illness [38]. The development of nanofungicides may offer some advantages, including increased fungicide efficacy and bioavailability, decreased toxicity, enhanced solubility of less water-soluble fungicides, targeted transport of the active ingredients, specific release, and extended shelf life. Among various kinds of nanoparticles and other kinds of nanomaterials, agro-nanofungicides, Z. multiflora, and ginger essential oil nanoformulations were shown to be effective and safe in the control of plant pathogenic fungi on a variety of crops. Nanotechnology has the potential to expand agricultural applications and alter the methods used now to manage plant pathogenic fungi. There is still a long way to go, but nanotechnology has paved the way towards a sustainable approach [315]. Agriculture is the basis of civilization—a promotion of the same can take mankind a long way into the future.

**Author Contributions:** Conceptualization, M.K.R. and Y.K.M.; software, S.M.; validation, I.C., N.A.K. and J.P.; resources, Y.K.M.; writing—original draft preparation, M.K.R., A.K.M. and Y.K.M.; writing—review and editing, I.C., N.A.K., S.K.A., J.P. and R.N.P.; figure preparation, S.M.; supervision, Y.K.M. and R.N.P. All authors have read and agreed to the published version of the manuscript.

**Funding:** This research received no external funding.

**Institutional Review Board Statement:** Not applicable.

**Data Availability Statement:** Not applicable.

**Acknowledgments:** The authors are highly indebted and extend their sincere thanks to SERB-DST, Government of India, for providing support to the Nano-biotechnology and Translational Knowledge Laboratory through Research Grant No. SRG/2022/000641, and support to RNP with Research Grant No. TAR/2020/000166.

**Conflicts of Interest:** The authors declare no conflict of interest.

**Abbreviations**

| | |
|---|---|
| Ag | Silver |
| Ag NW | Silver nanowire |
| Au | Gold |
| Au-MNPs | Magnetic gold nanoparticles |
| Au NP-ICTS | Gold nanoparticle-based immunochromatographic test strip |
| BBS | Bacterial brown stripe |
| BBSV | Broad bean stain virus |
| BBTV | Banana bunchy top virus |
| b-DNA | DNA with a bio-barcode |
| BYMV | Bean yellow mosaic virus |
| Cd | Cadmium |
| CdONP | Cadmium oxide nanoparticles |
| CdSe–PEI QD | Cadmium Selenium polyethylenimine–capped quantum dot |
| CdTe QD–CD | Cadmium telluride quantum dot |
| $CeO_2$ | Cerium oxide |
| CMV | Cucumber mosaic virus |
| CNTs | Carbon nanotubes |
| CS | Chitosan |
| CTV | Citrs tristeza virus |
| Cu | Copper |
| CuO | Copper oxide |
| CV | Cyclic voltammetry |
| Dot-ELISA | Dot enzyme-linked immunosorbent assay |
| DPV | Differential pulse voltammetry |
| EIS | Electrochemical impedance spectroscopy |
| ELISA | Enzyme-linked immunosorbent assay |
| E-nose | Electronic nose |
| $Fe_3O_4$ | Iron oxide |
| FFT-SWV | Fast Fourier transform square wave voltammetry |
| FISH | Fluorescence in situ hybridization |
| FRET | Fluorescence resonance energy transfer |
| GCE | Glassy carbon electrode |
| GO | Graphene oxide |
| GO-Ag | Graphene oxide-Silver |
| GST | glutathione-S-transferase |
| ITO | Indium-tin oxide |
| LDH | Lactate dehydrogenase |
| LFA | Lateral flow assay |
| LOD | Limit of detection |
| LPNE | Lithographically patterned nanowire electrodeposition |
| LSPR | Surface plasmon resonance |
| LSV | Linear sweep voltammetry |
| mAbs | Monoclonal antibodies |
| MCMV | Maize chlorotic mottle virus |
| MD | Mosaic disease |
| MgO | Magnesium oxide |
| MIP | Molecularly imprinted polymer |
| $MnO_2$ | Manganese dioxide |
| MTV | Tobacco mosaic virus |
| MWCNTs | Multiwalled carbon nanotubes |
| NGS | Next-generation sequencing |
| Ni | Nickel |
| NiO | Nickel oxide |
| NP | Nanoparticle |
| PCR | Polymerase chain reaction |
| Pd | Palladium |

| | |
|---|---|
| PGPR | Plant growth-promoting rhizobacteria |
| PPO | Polyphenol oxidase |
| PPY | Polypyrrole |
| PRSV | Papaya ringspot virus |
| PtNPs | Platinum nanoparticles |
| PVY | Potato virus Y |
| QDs | Quantum dots |
| RAPD | Random amplified polymorphic dna |
| rGO | Reduced graphene oxide |
| ROS | Reactive oxygen species |
| RTBV | Rice tungro bacilliform virus |
| RTSV | Rice tungro spherical virus |
| SAR | Systemic acquired resistance |
| Se | Selenium |
| SHRV | Sun hemp rosette virus |
| $SiO_2$ | Silicon oxide |
| SPCE | Screen-printed carbon electrode |
| SPR | Surface plasmon resonance |
| SRAP | Sequence-related amplified polymorphism |
| ssDNA | Single-strain deoxyribonucleic acid |
| SWCNTs | Single-walled carbon nanotubes |
| TBSV | Tomato bushy stunt virus |
| $TiO_2$ | Titanium oxide |
| TSP | Total soluble protein |
| TSWV | Tomato spotted wilt virus |
| TYLCV | Tomato yellow leaf curl virus |
| VOCs | Volatile organic compounds |
| WE SPCE | Working electrode screen-printed carbon electrode |
| YDV | Yellow dwarf virus |
| YMV | Yellow mosaic virus |
| ZnO | Zinc oxide |

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
