# Peer review of "Nanotechnology as a Promising Tool against Phytopathogens: A Futuristic Approach to Agriculture"

_agriculture, doi:10.3390/agriculture13091856_

Round 1
Reviewer 1 Report
The manuscript makes a relatively comprehensive review of the recent developments and future prospects of the application of nanopesticides on the control of phytopathogens. The English standard of the manuscript is in most of the cases sufficient. It has some academic value for scholars in the field of plant protection and material science. Although the topic is of great interest, the manuscript in its actual form is suffering from various issues that will need major revisions.
1. The layout of the article needs to be improved.
The topic should focus on corp disease control. However, the content enlarges to nanopesticides----not only fungicides, but also insecticides, herbicides. The author should delete content that is not relevant to disease control, or clarify the role of insecticides and herbicides in disease control.
2. Figures need to be improved。
Figure 2 “shelf-life is more” changed to “longer shelf-life”
“solubility is enhanced” to “enhanced solubility”
Figure 3 The first letter of the note in figure 2 must be in the same uppercase
“array based nanosensor”, “metal nanoparticles”, “nano barcoding”
Figure 5 The picture content is not closely related to the text content

The writing should conform to the standard of scientific paper and the language should be more precise.
Abbreviations should appear in the place where the words first appear, and they should be used in the following text consistently, instead of using full names and abbreviations alternatively (QD, ROS). Some abbreviations were used directly without listing their full names, which may cause difficulties in understanding. Besides, terms and abbreviations should be consistent in form and not be changed at will (nano fungicide or nano-fungicide). Kindly remind the authors to check the whole manuscript.
Author Response
- The layout of the article needs to be improved. The topic should focus on corp disease control. However, the content enlarges to nanopesticides----not only fungicides, but also insecticides, herbicides. The author should delete content that is not relevant to disease control, or clarify the role of insecticides and herbicides in disease control.
Author response: Thank you for nice suggestion. The content on nano insecticides and herbicide is removed as per suggestion.
- Figures need to be improved。
Figure 2 “shelf-life is more” changed to “longer shelf-life”
“solubility is enhanced” to “enhanced solubility”
Figure 3 The first letter of the note in figure 2 must be in the same uppercase
“array based nanosensor”, “metal nanoparticles”, “nano barcoding”
Figure 5 The picture content is not closely related to the text content.
Author response: The figures are modified as per suggestions.
- The writing should conform to the standard of scientific paper and the language should be more precise.
Author response: Thank you for the suggestion. The manuscript revised significantly, and all the discrepancies are removed.
4.Abbreviations should appear in the place where the words first appear, and they should be used in the following text consistently, instead of using full names and abbreviations alternatively (QD, ROS). Some abbreviations were used directly without listing their full names, which may cause difficulties in understanding. Besides, terms and abbreviations should be consistent in form and not be changed at will (nano fungicide or nano-fungicide). Kindly remind the authors to check the whole manuscript.
Author response: Thank you for the suggestion. The manuscript revised as per suggestions.
Reviewer 2 Report
Comments:
1. To date several similar review articles have been published. Therefore, authors should emphasis on the novelty of work.
2. As mentioned above, several articles are available in this fields, authors did not focus on any of such recent articles. No article found from 2023 and only 3-4 articles are there from 2022.
3. Considering, this it seems that authors are not really focused during preparation of this manuscript.
4. In addition, there are some scientific errors, e.g. scientific names of species should be italic
5. Authors should provide long form & specify short form in bracket when it is used first time in article.
6. There are some which needs re-phasing e.g. Page 14; Line 535-537: There is one extra ‘s’ in line number 537.
There are some typographical errors which needs to be rectified by authors.
There are some which needs re-phasing e.g. Page 14; Line 535-537: There is one extra ‘s’ in line number 537.
Author Response
- To date several similar review articles have been published. Therefore, authors should emphasis on the novelty of work.
Author response: The manuscript has included the all the holistic approaches that are involved in pathogen monitoring and control, and that have been emphasized in the revised manuscript.
- As mentioned above, several articles are available in these fields, authors did not focus on any of such recent articles. No article found from 2023 and only 3-4 articles are there from 2022.
Author response: Thank you for the suggestion. Recent studies are included in all possible aspects.
- Considering, this it seems that authors are not really focused during preparation of this manuscript.
Author response: Thank you for the suggestion. Recent studies are included in all possible aspects.
- In addition, there are some scientific errors, e.g. scientific names of species should be italic
Author response: Thank you for the suggestion. Necessary corrections are made.
- Authors should provide long form & specify short form in bracket when it is used first time in article.
Author response: Thank you for the suggestion. Necessary corrections are made.
- There are some which needs re-phasing e.g. Page 14; Line 535-537: There is one extra ‘s’ in line number 537.
There are some typographical errors which needs to be rectified by authors.
There are some which needs re-phasing e.g. Page 14; Line 535-537: There is one extra ‘s’ in line number 537.
Author response: Thank you for the suggestion. Necessary corrections are made in revised manuscript.
Reviewer 3 Report
The authors discuss the state of the art in using nanotechnology in diagnosis and therapy of plant diseases and their applications in agriculture. This is a very interesting topic and will have increasing impact in future. I liked to read the manuscript. But, I have some suggestions for its improvement:
I suggest to modifying the title: “Nanotechnology as promising tool box for detecting and controlling phytopathogens: A futuristic approach to agriculture”
Lines 61, 62: Nematodes are multicellular organisms. Beside them other non-vertebrate animals as insects can be responsible for important damages in plant cultures.
Line 132: for early investigations on antibiotics in agriculture, I recommend to cite: Köhler, Hedwig: Einführung in die Methoden der pflanzlichen Antibiotikaforschung, Akademie-Verlag, Berlin 1956 (https://de.wikipedia.org/wiki/Bakteriose)
Line 180: It would be fine to display a table with an overview on applications of nanoparticles in agriculture: plant disease/type of plants/type of nanoparticles/diagnosis/treatment/reference …
Lines 207 ..: the specific advantages and complementary use of special nanoparticles in comparison with genetic typization by PCR or DNA chips, functional genomics, antibody sensing should be explained, here.
Lines 222 -280 (section “Plant pathogen detection”): The subsections 4.1 – 4.5. are addressed to important strategies, but the possibilities and advantages should be described in more detail. If possible, comparisons with alternative techniques should be presented for 4.1. – 4.4
Lines 283-304 and lines 404-498 (nanoparticles in treatment of plant dieseaes): It would be nice to describe by a table which materials (lines 289-292) are used for special mechanisms (lines 294-304). Which pathogens are treated by the different types of nanomaterials? Are there preferences for the use of special nanoparticles for specific cultures? Tables would be very helpful to give the reader an overview on strategies and applications.
Lines 422…: The here described actions of nanoparticles are rather non-specific and bring, this way, non-desired environmental risks. For my opinion, nanotechnology should enable us to realize very specific activities against phytopathogens in order to avoid side damages. This should be discussed in relation to sections 5.1 – 5.3.
Lines 463, 464: Please discuss the environmental risks of heavy metal release from nanoparticles! What are the applied amounts/concentrations? What are the long-term consequences for soil and fertility-related soil microbial communities? How is the balance between profits and draw-backs in the application of such unspecific toxic materials?
Author Response
- I suggest to modifying the title: “Nanotechnology as promising tool box for detecting and controlling phytopathogens: A futuristic approach to agriculture”
Author response: Thank you for constructive suggestion with very catchy title. Modification has been made as suggested.
- Lines 61, 62: Nematodes are multicellular organisms. Beside them other non-vertebrate animals as insects can be responsible for important damages in plant cultures.
Author response: Corrected.
- Line 132: for early investigations on antibiotics in agriculture, I recommend to cite: Köhler, Hedwig: Einführung in die Methoden der pflanzlichen Antibiotikaforschung, Akademie-Verlag, Berlin 1956 (https://de.wikipedia.org/wiki/Bakteriose)
Author response: Cited in Section 4.2.
- Line 180: It would be fine to display a table with an overview on applications of nanoparticles in agriculture: plant disease/type of plants/type of nanoparticles/diagnosis/treatment/reference …
Author response: Thanks to the reviewer for the constructive suggestion. All points raised by the reviewers are addressed in Section 5, Table 2 and Figure 5.
- Lines 207 ..: the specific advantages and complementary use of special nanoparticles in comparison with genetic typization by PCR or DNA chips, functional genomics, antibody sensing should be explained, here.
Author response: A small paragraph has been placed in the Section 3 with appropriate reference.
- Lines 222 -280 (section “Plant pathogen detection”): The subsections 4.1 – 4.5. are addressed to important strategies, but the possibilities and advantages should be described in more detail. If possible, comparisons with alternative techniques should be presented for 4.1. – 4.4
Author response: Thank you for the critical suggestion. That section 4.1-4.5 has been modified as Section 3 and all points are addressed there. One comprehensive table has been also given for better understanding to the audience.
- Lines 283-304 and lines 404-498 (nanoparticles in treatment of plant dieseaes): It would be nice to describe by a table which materials (lines 289-292) are used for special mechanisms (lines 294-304). Which pathogens are treated by the different types of nanomaterials? Are there preferences for the use of special nanoparticles for specific cultures? Tables would be very helpful to give the reader an overview on strategies and applications.
Author response: Thanks to the reviewer for the constructive suggestion. All points raised by the reviewers are addressed in Section 5, Table 2 and Figure 5. Section 6 presents the details mechanisms with most recent references.
- Lines 422…: The here described actions of nanoparticles are rather non-specific and bring, this way, non-desired environmental risks. For my opinion, nanotechnology should enable us to realize very specific activities against phytopathogens in order to avoid side damages. This should be discussed in relation to sections 5.1 – 5.3.
Author response: Thank you for the suggestion regards to address a critical point during application of nanoparticles in to the field. Authors are really grateful. The details are mentioned with most recent works in Section 7, Table 3 and Figure 7.
- Lines 463, 464: Please discuss the environmental risks of heavy metal release from nanoparticles! What are the applied amounts/concentrations? What are the long-term consequences for soil and fertility-related soil microbial communities? How is the balance between profits and draw-backs in the application of such unspecific toxic materials?
Author response: Thank you for the suggestion regards to address a critical point during application of nanoparticles in to the field. Authors are really grateful. The details are mentioned with most recent works in Section 7, Table 3 and Figure 7.
Round 2
Reviewer 1 Report
The authors have made great effort to improve the manuscript.
The language has been improved.
Author Response
No comments from reviewer
Reviewer 2 Report
No further comments
Author Response
No comments by reviewer
Reviewer 3 Report
The authors answered to all points. The only further required correction concerns the citation ref125: the author is Hedwig Köhler:
Hedwig Köhler:
Einführung in die Methoden der pflanzlichen Antibiotikaforschung
Deutsche Akademie der landwirtschaftswissenschaften zu Berlin, Wissenschaftliche Abhandlungen Nr 13, Akademie-Verlag, Berlin, 1956.
The contribution of J. Grosjean is a review on this book, only: (Grosjean, J. Einführung in Die Methoden Der Pflanzlichen Antibiotikaforschung. Tijdschr. Over Planteziekten 1957, 63, 159– 1324
159, doi:10.1007/bf01980695. )
Author Response
Reference corrected as per suggestion